# Towards Efficient 3D Object Detection with Knowledge Distillation

**Jihan Yang[1]    Shaoshuai Shi[2]    Runyu Ding[1]    Zhe Wang[3]    Xiaojuan Qi[1]**

[1]The University of Hong Kong    [2]Max Planck Institute for Informatics    [3]SenseTime Research
{jhyang, ryding, xjqi}@eee.hku.hk, {shaoshuaics, wzlewis16}@gmail.com

## Abstract

Despite substantial progress in 3D object detection, advanced 3D detectors often suffer from heavy computation overheads. To this end, we explore the potential of knowledge distillation (KD) for developing efficient 3D object detectors, focusing on popular pillar- and voxel-based detectors. In the absence of well-developed teacher-student pairs, we first study how to obtain student models with good trade offs between accuracy and efficiency from the perspectives of model compression and input resolution reduction. Then, we build a benchmark to assess existing KD methods developed in the 2D domain for 3D object detection upon six well-constructed teacher-student pairs. Further, we propose an improved KD pipeline incorporating an enhanced logit KD method that performs KD on only a few pivotal positions determined by teacher classification response, and a teacher-guided student model initialization to facilitate transferring teacher model's feature extraction ability to students through weight inheritance. Finally, we conduct extensive experiments on the Waymo and KITTI dataset. Our best performing model achieves $65.75\%$ LEVEL 2 mAPH, surpassing its teacher model and requiring only $44\%$ of teacher flops on Waymo. Our most efficient model runs 51 FPS on an NVIDIA A100, which is $2.2\times$ faster than PointPillar with even higher accuracy on Waymo. Code is available at https://github.com/CVMI-Lab/SparseKD.

## 1   Introduction

3D object detection from point clouds is a fundamental perception task with broad applications on autonomous driving, robotics and smart city, etc. Recently, benefited from large-scale 3D perception datasets [9, 2, 37] and advanced point- [29], pillar- [21, 43] and voxel-based [10, 51] representations of sparse and irregular LiDAR point cloud scenes, 3D detection has achieved remarkable progress [46, 35, 33, 1, 49]. However, stronger performance is often accompanied with heavier computation burden (see Figure 1), rendering their adoption in real-world applications still a challenging problem.

Recent attempts to improve efficiency focus on developing specified architectures for point-based 3D object detectors [5, 50], not generalizable to a wide spectrum of pillar/voxel-based methods [51, 21, 46, 33, 49, 7]. Here, we aim at a model-agnostic framework for obtaining efficient and accurate 3D object detectors with

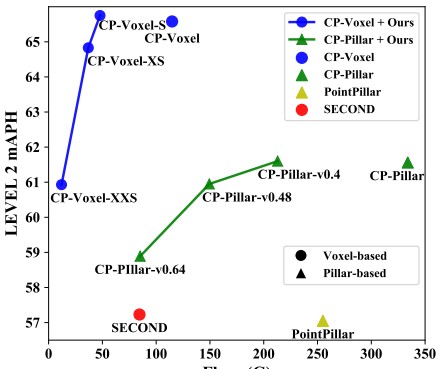

Figure 1: Performance and flops comparison of single-stage detectors on Waymo: CP-Pillar [49], CP-Voxel [49], SECOND [46], PointPillar [21] and Ours.

knowledge distillation (KD). Due to its effectiveness, generality and simplicity, KD has become a de facto strategy to develop efficient models in a variety of 2D tasks [15, 24, 6, 16, 45]. It facilitates

36th Conference on Neural Information Processing Systems (NeurIPS 2022).

improving the performance of a lightweight and efficient student model by harvesting knowledge learned by an accurate yet computationally heavy teacher model. Despite of numerous studies in 2D tasks, the investigation of KD for efficient 3D object detection has largely escaped research attention with unresolved research challenges. In this paper, we conduct the first systematic study on knowledge distillation for developing high-performance and efficient 3D LiDAR-based detectors.

First, we study *how to obtain lightweight student detectors with satisfactory efficiency and accuracy trade offs given a pre-trained teacher 3D object detector.* Different from the 2D domain where well-developed backbone architectures with different model efficiencies are readily available (e.g., ResNet 18 *vs.* ResNet 50) [36, 39, 13], such scalable backbones in 3D scenes are still under-explored. This makes the design of suitable student models a non-trivial problem. Intuitively, a good student model should achieve a good compromise between accuracy and efficiency as a poor student model may have inferior capabilities or architectural level shortcomings, causing difficulties for further knowledge distillation. In this perspective, we first propose the Cost Performance Ratio (CPR) to fairly evaluate a model in terms of both efficiency and capability. Then, we study different factors including model width (*i.e.* number of channels), depth (*i.e.* number of layers) and input resolution on both pillar-based and voxel-based architectures. Specifically, we find that pillar-based architecture favors input-level compression (*i.e.* reduce input resolution) while voxel-based detector prefers width-level compression, due to less spatial redundancy of voxel-based detectors.

Second, we empirically investigate *the effectiveness of existing knowledge distillation methods on this new setting* upon accurate teacher models and efficient student models. As there is no prior research for this problem, we benchmark seven existing knowledge distillation methods on top of six teacher-student pairs covering voxel- and pillar-based architectures. Specifically, we evaluate the following major streams: **logit KD** distilling on model outputs (KD [15] and GID-L [6]), **feature KD** mimicking intermediate features (FitNet [32], Mimic [22], FG [42] and GID-F [6]) and **label KD** leveraging teacher's predictions for label assignment [28]. Further, we also study their synergy effects and empirically find that feature KD outperforms others individually, but fails to cooperate with other KD manners to further boost the performance in 3D detection. This is potentially caused by the fact that logit and label KD can add an implicit regularization to the intermediate features. The best strategy that we obtain through empirical studies is to use FG [42] or Mimic [22] individually.

Third, we propose *simple, general, and effective strategies to improve knowledge distillation on 3D object detection* upon the strong KD baseline derived as above. Motivated by the extreme imbalance between small informative areas containing 3D objects and large redundant background areas in 3D scenes, we design a modified logit KD method, namely pivotal position logit KD, enforcing imitation on only locations with highly confident or top-ranked teacher predictions. These areas are shown to be near instance centers or error-prone positions. Besides, to facilitate effective knowledge transfer from teacher to student, we develop Teacher Guided Initialization (TGI) which remaps pre-trained teacher parameters to initialize a student model. This is shown to be effective in inheriting teacher models' feature extraction abilities while collaborating well with logit and label KD techniques.

Finally, our empirical studies on efficient model design and knowledge distillation methods yield superior performance in delivering efficient and effective pillar- and voxel-based 3D detectors. This is extensively verified on the largest annotated 3D object detection dataset – Waymo [37]. As shown in Figure 1, our best performing model, *i.e.* CP-Voxel-S, even outperforms its teacher model (*i.e.* CP-Voxel) while has $2.4\times$ fewer flops. Moreover, our most efficient pillar-based model (*i.e.* CP-Pillar-v0.64) can run 51 FPS with $58.89\%$ mAPH while previous fastest voxel/pillar-based detector – PointPillar runs 23 FPS with $57.03\%$ mAPH on an NVIDIA A100 GPU (see Suppl.). Our method is also shown to be effective in knowledge transfer from heavy two-stage object detectors to lightweight single-stage detectors. In addition, our method can generalize well to other settings such as KITTI dataset with SECOND as well as advance compression methods in Sec. 5.5, other detectors, and even 3D semantic segmentation in Suppl..

## 2 Related Work

**3D LiDAR-based Object Detection** targets to localize and classify 3D objects from point clouds. Point-based methods [35, 5, 47, 50] took raw point clouds and leveraged PointNet++ [29] to extract sparse point features and generate point-wise 3D proposals. Pillar-based works [21, 43] finished voxelization in bird eye's view and extracted pillar-wise features with PointNet++. Voxel-based methods [51, 46, 33] voxelized point clouds and obtained voxel-wise features with 3D sparse convolutional networks, which is the most popular data treatment. Besides, range-based works [1, 38]

were proposed for long-range and fast detection. Recently, designing efficient 3D detectors has drawn some attentions [5, 50] with raw point data treatment. In this work, we focus on exploring model-agnostic knowledge distillation methods to boost the efficiency of 3D detectors.

**Knowledge Distillation** aims to transfer knowledge from a large teacher model to a lightweight student network, which is a thriving area in efficient deep learning. Hinton *et al.* proposed the seminal concept of knowledge distillation (KD) [15], which distilled knowledge between teacher and student on the output level (*i.e.* prediction logits). Another line of research proposed to help student's optimization with hints stored in informative intermediate features from teacher [32, 17, 20, 14, 19, 4]. In addition, some works attempted distillation techniques in 2D object detection [22, 42, 6, 30, 48, 28] by emphasizing instance-wise distillation and feature knowledge. Mimic [22], FG [42] and GID [6] sampled local region features with box proposals or custom indicators for foreground-aware feature imitation. Label KD [28] utilized teacher's information for label assignment of student. Recently, knowledge distillation has also been leveraged to transfer knowledge in multi-modality setup [11, 25] or multi-frame to single-frame setup [44] in 3D detection area. However, to the best of our knowledge, we are the first to explore knowledge distillation in the most popular setup: single-frame 3D LiDAR-based object detection. In this work, we propose an enhanced 3D detection KD pipeline with our designed efficient 3D detectors on the popular voxel/pillar data representation. Our lightweight detectors CP-Voxel-S and CP-Pillar-v0.4 slightly outperform their state-of-the-art teacher detectors separately while requiring much less computation overhead.

# 3 Designing Efficient Student Networks

As there are no readily available lightweight backbone architectures for constructing student networks, we carry out an empirical study on how to obtain an efficient model with satisfactory efficiency and accuracy trade offs to facilitate further knowledge distillation. In this section, we will first describe our experimental setups and model evaluation metrics. Then, we study different strategies to obtain an efficient model and conduct an in-depth analysis on how to achieve good trade offs for pillar- and voxel-based architectures.

## 3.1 Basic Setups and Evaluation Metrics

**Basic setups.** For detector architectures, we focus on two variants of the state-of-the-art model CenterPoint [49]: CenterPoint-Pillar (CP-Pillar) and CenterPoint-Voxel (CP-Voxel), covering the most popular pillar- and voxel-based 3D detectors [46, 51, 21, 43, 33, 34, 7]. For dataset, we perform all experiments on the largest annotated 3D LiDAR perception dataset Waymo Open Dataset (WOD) [37] with $20\%$ training samples for fast verification. For model training, we follow the training scheme of popular 3D detection codebase OpenPCDet [41] to ensure fair comparisons and standardization. Note that there is no any knowledge distillation method engaged in this stage.

**Evaluation metrics.** Following [31, 23], we employ number of parameters, flops, activations, latency (*i.e.* test time) and peak GPU training memory (batch size 1) as quantitative indicators to evaluate model efficiency from parameter, computation and memory throughout aspects. Note that activations, flops and latency are averaged over 99 frames with a GTX-1060 GPU. We present latency more for reference since it largely depends on hardware devices as well as operation-level optimizations (see Sec. **??**). We use LEVEL 2 mAPH as the performance evaluation metric following WOD [37].

Since our major target here is to design student networks with favorable trade offs between performance and efficiency, we propose a quantitative indicator, namely Cost Performance Ratio (CPR), to directly measure a model in this respective. To construct CPR, we use activations (acts) as the metric to evaluate efficiency since it strongly correlates with the runtime on hardware accelerators such as GPUs as shown in [31]. Our CPR finally combines the activation ratio and mAPH as follows:

$$\text{CPR} = 0.5 \times (1 - \frac{\text{acts}_s}{\text{acts}_t}) + 0.5 \times (\frac{\text{mAPH}_s}{\text{mAPH}_t})^3, \tag{1}$$

where the subscripts $s$ and $t$ represent student and teacher model respectively. CPR is normalized to $[0, 1]$ by weighting the relative activation decrease and performance drop ratio of a student network compared to the teacher. Notice that the third power is used for the performance degradation term to penalize acceleration methods that result in drastic performance degradation more severely. We argue that it is necessary to ensure a relative good performance of efficient detectors as models with poor accuracy may suffer from architectural level problems which can cause difficulties in developing knowledge distillation techniques and prohibiting obtaining accurate and efficient detectors.

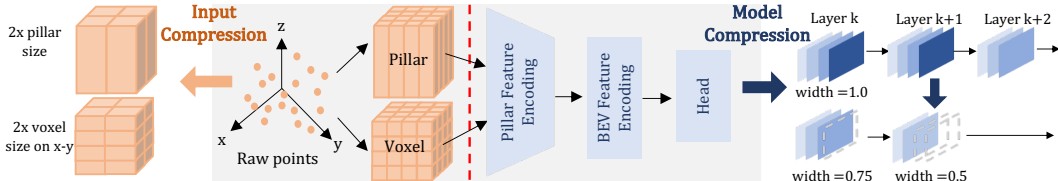

Figure 2: Architecture of detector and illustration of model (right) and input (left) compression.

## 3.2 Acceleration Strategy

Here, to obtain efficient models, we investigate model and input resolution compression techniques on pillar- and voxel-based 3D detectors as shown in Figure 2 and study their impacts on model accuracy.

**Model Compression.** Here, we compress a given teacher backbone by trimming it along depth (*i.e.* number of layers) or width (number of channels) as shown in the right of Figure 2. To relieve the burden of exhaustive layer-wise studies, we leverage the functional similarity among different layers to group them into three major modules: Pillar Feature Encoding (PFE) module, Bird eye's view Feature Encoding (BFE) module and detection head (see Figure 2: right), and conduct analysis on module-level. Specifically, PFE corresponds to network components before projecting features to bird eye's view (BEV) grid, including sparse 3D convolutional backbone [10] to extract voxel-wise features in voxel-based architectures or PointNet++ [29] to encode pillar-wise features from points in pillar-based detectors. After PFE, the features are aggregated to pillar features and mapped to 2D BEV grid. Then, the BFE consisting of 2D convolutional layers is adopted to extract final detection feature map on the BEV grid. Finally, the detection head takes outputs from BFE to produce final prediction results. For width pruning, we consider the width of teacher detectors as 1.0, and slim each module of the student by decreasing its number of channels with a given width. Depth trimming also follows this paradigm but still maintains the minimal structure of each module (*e.g.* downsample and upsample layers) to ensure basic detection ability. Model trimming results are shown in Table 1.

Table 1: Model compression results. Teacher models are marked in gray. See text for details.

| Architecture | | | | | | Efficiency | | | | | LEVEL 2 mAPH | CPR |
| Detector | Width | | | Depth | | Params (M) | Flops (G) | Acts (M) | Latency (ms) | Mem. (G) | | |
| | PFE | BFE | Head | PFE | BFE | | | | | | | |
|---|---|---|---|---|---|---|---|---|---|---|---|---|
| CP-Pillar | 1.00 | 1.00 | 1.00 | 1.00 | 1.00 | 5.2 | 333.9 | 303.0 | 157.9 | 5.2 | 59.09 | - |
| (a) | 1.00 | 0.50 | 0.50 | 1.00 | 1.00 | 1.3 | 87.6 | 161.8 | 78.5 | 3.2 | 54.50 | 0.63 |
| (b) | 0.50 | 0.50 | 0.50 | 1.00 | 1.00 | 1.3 | 85.0 | 152.7 | 74.0 | 2.9 | 52.33 | 0.60 |
| (c) | 1.00 | 1.00 | 1.00 | 1.00 | 0.50 | 2.2 | 258.5 | 234.1 | 118.5 | 4.3 | 55.24 | 0.52 |
| (d) | 1.00 | 1.00 | 1.00 | 1.00 | 0.33 | 1.4 | 234.6 | 210.0 | 107.8 | 4.0 | 47.97 | 0.42 |
| CP-Voxel | 1.00 | 1.00 | 1.00 | 1.00 | 1.00 | 7.8 | 114.7 | 101.9 | 125.7 | 2.8 | 64.29 | - |
| (a) | 1.00 | 0.50 | 0.50 | 1.00 | 1.00 | 4.0 | 47.8 | 65.7 | 98.0 | 2.1 | 62.23 | 0.63 |
| (b) | 0.75 | 0.50 | 0.50 | 1.00 | 1.00 | 2.8 | 36.9 | 58.4 | 88.2 | 1.9 | 61.16 | 0.64 |
| (c) | 0.50 | 0.50 | 0.50 | 1.00 | 1.00 | 1.9 | 28.8 | 51.2 | 75.1 | 1.7 | 59.47 | 0.64 |
| (d) | 0.50 | 0.25 | 0.25 | 1.00 | 1.00 | 1.0 | 12.0 | 33.1 | 70.4 | 1.3 | 56.26 | 0.67 |
| (e) | 0.25 | 0.25 | 0.25 | 1.00 | 1.00 | 0.5 | 7.3 | 25.8 | 66.0 | 1.1 | 49.84 | 0.61 |
| (f) | 1.00 | 1.00 | 1.00 | 0.50 | 0.50 | 3.0 | 63.9 | 65.2 | 73.0 | 1.9 | 60.95 | 0.61 |
| (g) | 1.00 | 1.00 | 1.00 | 0.33 | 0.33 | 1.8 | 47.9 | 52.2 | 59.0 | 1.6 | 55.78 | 0.57 |

**Input Compression.** Besides model complexities, input resolution also has impacts on model efficiency [40, 30]. For instance, by halving the input resolution, the computation overhead for 2D convolution layers in the BFE module and detection head will be reduced to $\frac{1}{4}$ (see Figure 2: left). Besides, there are large background areas in the sparse and large-scale 3D scenes, which naturally has redundancies and offers the possibility of processing the data on a coarser resolution for input compression. Specifically, input compression is realized by increasing the voxel/pillar size on the x-y plane when constructing voxel/pillar. As shown in Table 2, we gradually increase students' voxel size with 25% of teachers' voxel size, and record their efficiency and accuracy metrics.

## 3.3 Conclusion and Analysis

By analyzing the experimental results shown in Table 1 and Table 2, we draw the following conclusions on efficient model design for pillar- and voxel-based detectors.

**Width *vs.* depth compression: width-level pruning is preferred.** As shown in Table 1, trimming networks on width generally achieves higher CPR than on depth for both CP-Pillar and CP-Voxel. For instance, with stronger performance, CP-Voxel (d) needs $1.5\times$ fewer acts and $4\times$ fewer flops compared to CP-Voxel (g). As backbones of 3D detectors are much shallower than their 2D counterparts

Table 2: Input compression results. Teacher models are marked in gray. See text for details.

| Architecture | | Efficiency | | | | | LEVEL 2 | CPR |
|---|---|---|---|---|---|---|---|---|
| Detector | Voxel Size (m) | Params (M) | Flops (G) | Acts (M) | Latency (ms) | Mem. (G) | mAPH | |
| CP-Pillar | 0.32 | 5.2 | 333.9 | 303.0 | 157.9 | 5.2 | 59.09 | - |
| | 0.40 | 5.2 | 212.9 | 197.7 | 103.4 | 3.8 | 57.55 | 0.64 |
| | 0.48 | 5.2 | 149.4 | 142.3 | 81.9 | 3.0 | 56.27 | 0.70 |
| | 0.56 | 5.2 | 109.9 | 109.0 | 66.3 | 2.6 | 54.45 | 0.71 |
| | 0.64 | 5.2 | 85.1 | 88.0 | 54.5 | 2.1 | 52.81 | 0.71 |
| CP-Voxel | 0.100 | 7.8 | 114.8 | 101.9 | 125.7 | 2.8 | 64.29 | - |
| | 0.125 | 7.8 | 77.5 | 70.1 | 99.9 | 2.2 | 61.55 | 0.59 |
| | 0.150 | 7.8 | 53.9 | 50.0 | 84.3 | 1.8 | 58.14 | 0.62 |
| | 0.175 | 7.8 | 44.3 | 41.2 | 74.1 | 1.5 | 55.99 | 0.63 |
| | 0.200 | 7.8 | 32.9 | 31.4 | 67.5 | 1.3 | 52.80 | 0.62 |

(*e.g.* only 19 convolution layers in CP-Pillar and 35 convolution layers in CP-Voxel), this renders depth compression more challenging and less scalable than width compression in 3D detection.

**Module-wise pruning selection: PFE module has the least redundancy to be reduced.** Since PFE, BFE and detection head perform different functions, they might have their own redundancies in 3D detection. Comparing CP-Voxel (c), (d) and (e) as well as CP-Pillar (a) and (b) in Table 1, we find that less pruning on PFE achieves significantly higher CPR, demonstrating that network parameters in the PFE module is crucial for high performance and it is necessary to maintain network complexities in PFE modules.

**Favorable compression strategies for different detection architectures.** Comparing results of different compression strategies in Table 1 and 2, we find that pillar-based architecture (*i.e.* CP-Pillar) is more suitable for input compression while voxel-based architecture (*i.e.* CP-Voxel) prefers width-level compression. This difference mainly lies in different spatial redundancy of BEV features on these two detectors. Since the PFE module of voxel-based detector downsamples input resolution to $1/8$, the $0.1m$ input voxel size will be magnified to $0.8m$ on BEV features. Hence, less spatial redundancy could be further compressed for CP-Voxel. However, CP-Pillar often keeps the same resolution between input and BEV features on Waymo [41, 49], and thus has larger resolution redundancy on BEV grid, which facilitates designing student network with coarser input resolution.

**Summarized student networks.** Driven by the above analysis, we finally derive the following compressed student models with good trade offs between efficiency and performance. For CP-Pillar, as it is friendly to input compression, we adopt the input-compressed models with voxel size 0.40, 0.48 and 0.64 in Table 2, named **CP-Pillar-v0.4**, **CP-Pillar-v0.48** and **CP-Pillar-v0.64**, respectively. As for CP-Voxel, we choose its width compressed models (a) (b) and (d) in Table 1, named **CP-Voxel-S**, **CP-Voxel-XS** and **CP-Voxel-XXS**, respectively.

## 4 Benchmark Knowledge Distillation for 3D Object Detection

With student networks derived in Section 3, we are now ready to conduct our empirical study on knowledge distillation for 3D object detection. Here, we benchmark seven popular 2D KD methods including logit KD (*i.e.* KD [15] and GID-L [6]), feature KD (*i.e.* FitNet [32], Mimic [22], FG [42] and GID-F [6]) and label KD [28] on six teacher-student pairs with comprehensive analysis. Notice that GID and label KD are state-of-the-art 2D KD detection methods, and GID is divided into GID-L and GID-F to investigate logit KD and feature KD separately. Implementation details and the value of hyper-parameters are described in the Suppl..

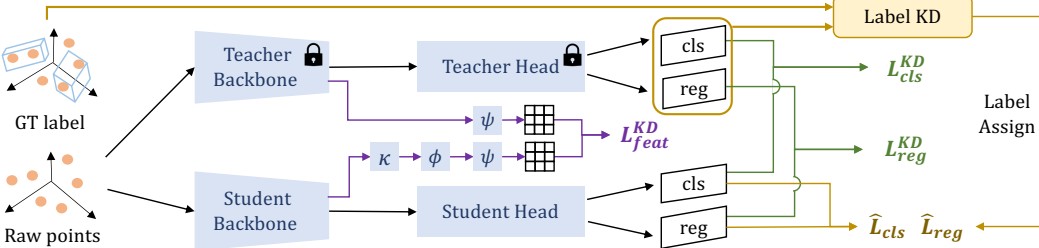

Figure 3: Overall KD paradigm for 3D detection. Teacher weights are frozen during the whole distillation procedure. Logit, feature and label KD are colored by green, purple and yellow, respectively.

## 4.1 Paradigm

We first evaluate existing 2D distillation methods for 3D detection. As shown in Figure 3, our overall KD paradigm for 3D detection contains three parts: logit, feature and label KD to leverage teacher guidance in response, intermediate feature and label assignment levels, respectively.

**Logit KD** is the most classical distillation approach introduced by [15]. It takes teacher model's final response as guidance for training a student network and is closely related to the specific task. In 3D object detection, we calculate the logit KD loss between teacher and student outputs as follows:

$$\mathcal{L}_{\text{cls}}^{\text{KD}} = \mathbb{E}[m_{\text{cls}}\|\kappa(p_{\text{cls}}^s) - p_{\text{cls}}^t\|_2], \qquad \mathcal{L}_{\text{reg}}^{\text{KD}} = \mathcal{L}_{\text{reg}}(p_{\text{reg}}^s, \ p_{\text{reg}}^t), \qquad (2)$$

where superscripts $s$ and $t$ indicate student and teacher, $p_{\text{cls}}$ and $p_{\text{reg}}$ represent the classification response after the sigmoid and bounding box regression prediction of detector separately, $\kappa$ is the bilinear interpolation to match student output resolutions towards teacher, $\mathcal{L}_{\text{reg}}$ is the regression loss function of 3D detector and $m_{\text{cls}}$ is a mask ranged in $[0, 1]$ to indicate important regions in $p_{\text{cls}}$ (see green parts in Figure 3). Compared to vanilla KD [15] which mimics all teacher outputs, GID-L only focuses on some local regions covered by selected box proposals and results in different $m_{\text{cls}}$.

**Feature KD** is the major stream of work in KD for 2D object detection. It enforces student models to mimic teacher models' intermediate feature maps. Specifically, we construct feature mimicking on the last layer of BFE between student and teacher network as follows:

$$\mathcal{L}_{\text{feat}}^{\text{KD}} = \mathbb{E}[m_{\text{feat}}\|\psi(\phi(\kappa(f^s)), y) - \psi(f^t, y)\|_2], \qquad (3)$$

where $y$ is the ground truth, $\psi$ indicates the RoI Align [12], $\phi$ is a $1 \times 1$ convolution with batch normalization [18] and ReLU [27] block to align channel-wise discrepancy between teacher feature $f^t$ and student feature $f^s$, $m_{\text{feat}}$ is the mask to indicate critical regions ranged in $[0, 1]$ (see purple parts in Figure 3). Considering the imbalance between foreground and background regions in the detection, a prevailing solution is to emphasize near object regions to perform distillation [22, 42, 6]. Different investigated feature KD methods mainly vary in the selection of such critical regions $m_{\text{feat}}$.

**Label KD** is a newly proposed distillation strategy, which leverages teacher predictions in the label assignment stage of student [28]. Motivated by the simple and general label KD, we also employ it as a KD baseline method. Specifically, given a point cloud scene $x$ and its corresponding ground truth (GT) set $y$, label KD first obtains the prediction $y^t$ and confidence score $o^t$ from the pretrained teacher detector. After filtering $o^t$ with a given threshold $\tau$, it then obtains a high-quality teacher prediction set $\hat{y}^t$. It generates a teacher assisted GT set $\hat{y}^{\text{KD}} = \{y, \hat{y}^t\}$ by combining GTs as well as confident teacher predictions and carries on label assignment for student with $\hat{y}^{\text{KD}}$. The final classification and regression losses with label KD on student detectors are $\hat{\mathcal{L}}_{\text{cls}}$ and $\hat{\mathcal{L}}_{\text{reg}}$.

**Training Objective** is the combination of three stream KD techniques as follows:

$$\mathcal{L} = \hat{\mathcal{L}}_{\text{cls}} + \lambda\hat{\mathcal{L}}_{\text{reg}} + \alpha_1\mathcal{L}_{\text{cls}}^{\text{KD}} + \alpha_2\mathcal{L}_{\text{reg}}^{\text{KD}} + \alpha_3\mathcal{L}_{\text{feat}}^{\text{KD}}, \qquad (4)$$

where $\lambda$, $\alpha_1$, $\alpha_2$ and $\alpha_3$ are trade-off parameters between different objectives. Note that the existence and implementation of each operation (*e.g.* $\psi$, $\kappa$, $m_{\text{feat}}$, $m_{\text{cls}}$, etc) varies among different KD methods.

Table 3: Knowledge distillation benchmark for 3D detection on Waymo. Performance are measured in LEVEL 2 mAPH. Best and second-best methods are noted by **bold** and underline, respectively. "Ours" indicates our proposed improved knowledge distillation method introduced in Sec. 5

| Detector | No Distill | Logit KD | | Feature KD | | | | Label KD | Ours | Flops (G) | Acts (M) |
|---|---|---|---|---|---|---|---|---|---|---|---|
| | | KD | GID-L | FitNet | Mimic | FG | GID-F | KD | | | |
| CP-Pillar | 59.09 | - | - | - | - | - | - | - | - | 333.9 | 303.0 |
| CP-Pillar-v0.4 | 57.55 | 57.51 | 57.54 | 57.89 | 58.57 | 58.44 | 58.26 | 58.10 | **59.24** | 212.9 | 197.7 |
| CP-Pillar-v0.48 | 56.27 | 55.76 | 56.29 | 55.82 | 57.26 | 57.26 | 57.23 | 57.54 | **58.53** | 149.4 | 142.3 |
| CP-Pillar-v0.64 | 52.81 | 53.13 | 50.78 | 51.79 | 53.83 | 53.37 | 53.18 | 53.78 | **55.82** | 85.1 | 88.0 |
| CP-Voxel | 64.29 | - | - | - | - | - | - | - | - | 114.7 | 101.9 |
| CP-Voxel-S | 62.23 | 62.81 | 62.89 | 60.51 | 63.35 | 63.33 | 62.75 | 63.31 | **64.25** | 47.8 | 65.7 |
| CP-Voxel-XS | 61.16 | 61.30 | 62.25 | 58.94 | 62.23 | 62.48 | 62.42 | 61.81 | **63.53** | 36.9 | 58.4 |
| CP-Voxel-XXS | 56.26 | 56.11 | 57.19 | 52.24 | 57.00 | 57.92 | 57.16 | 57.02 | **59.28** | 12.0 | 33.1 |

## 4.2 Results and Analysis

**Benchmark Analysis.** As shown in Table 3, compared to the no distillation baseline, all three streams of KD methods obtain performance improvements on six teacher-student pairs. Among seven KD baseline strategies, feature-based KD methods (*i.e.* Mimic and FG) achieve prominent performance, which demonstrates the strong potential of learning from teacher's hints on feature

extraction. Furthermore, we find that instance-aware local region imitation is important in distillation for 3D detection, as enormous background regions overwhelm the supervision of sparse instances. For instance, with instance-aware imitation, Mimic, FG and GID-F consistently outperform FitNet which fully imitates all spatial positions of teacher feature maps. Similar conclusions can also be drawn in logit KD by comparing the results of instance-aware GID-L and vanilla KD.

**Synergy Analysis.** While benchmark results in Table 3 mainly focus on the individual effectiveness of each KD manner, their synergy effect is also an important consideration, which has the potential to further improve student performance. As shown in Table 4, although feature KD itself achieves the highest performance on CP-Voxel-XXS compared to logit KD and label KD techniques, it can hardly achieve improvements or even suffers from performance degradation when combined with other KD methods. On the contrary, logit KD and label KD can cooperate well with each other to further improve student's capability. This is potentially caused by logit KD and label KD implicitly enforcing regularization on the feature, which can be conflicted with the optimization direction of feature KD and results in a poor synergy effect.

Table 4: Synergy investigation based on CP-Voxel-XXS.

| GID-L | Label KD | FG | mAPH |
|---|---|---|---|
| | | | 56.26 |
| √ | | | 57.19 |
| | √ | | 57.02 |
| | | √ | 57.92 |
| √ | √ | | 57.60 |
| √ | | √ | **58.01** |
| | √ | √ | 57.06 |
| √ | √ | √ | 57.62 |

## 5 Improved Knowledge Distillation for 3D Object Detection

As shown in Table 4, the basic knowledge distillation pipeline fails to achieve remarkable results by combining the best method of three KD streams (*i.e.* GID-L [6], label KD [28] and FG [42]). In the following, we propose an improved KD pipeline for 3D object detection, including pivotal position logit KD to alleviate the extreme imbalance between foreground and background regions through only imitating response on sparse pivotal positions, label KD, and teacher guided initialization scheme to further facilitate transferring teacher's feature extraction ability to student models.

### 5.1 Method

**Pivotal Position Logit KD.** Motivated by the imbalance of foreground and background regions, previous 2D methods attempt to only enforce output-level imitation on pixels near or covering instances [42, 6]. However, we find that it is sub-optimal in 3D scenarios given more extreme imbalance between small informative instances and large redundant background areas. For example, based on CP-Pillar, even a vehicle with $10m$ length and $4m$ width occupies only $32 \times 13$ pixels in the final $468 \times 468$ response map. Such small instances and large perception ranges in 3D detection requires more sophisticated imitation region selection than previous coarse instance-wise masking manners in 2D detection. Hence, we propose Pivotal Position (PP) logit KD which leverages cues in teacher classification response or label assignment to determine the important areas for distillation.

Specifically, pivotal position selection can be formulated as finding suitable $m_{\text{cls}}$ in Eq. (2). Here, we show three variants to obtain it. First, confidence of high-performance teacher prediction can serve as a valuable indicator to figure out pivotal positions for student (*i.e.* confidence PP). By filtering teacher confidence $o^t$ with a threshold $\tau_{pp}$, we then set $i, j$ positions of $m_{\text{cls}}$ with $o^t_{i,j} \geq \tau_{pp}$ to one and otherwise zero. With similar spirit, we can also select top-ranked $K$ positions (*i.e.* rank PP) in teacher classification response $P^t_{\text{cls}}$ as pivotal positions and convert them to a one-hot embedded $m_{\text{cls}}$. These confident or top-ranked positions are shown to be near object centers or error-prone regions. Last, inspired by the Gaussian label assignment in CenterPoint [49], we can define pivotal positions in a soft way with center-peak Gaussian distribution for each instance as $m_{\text{cls}}$ (*i.e.* Gaussian PP). We empirically show that all three variants of PP logit KD achieve promising gains in 3D detection.

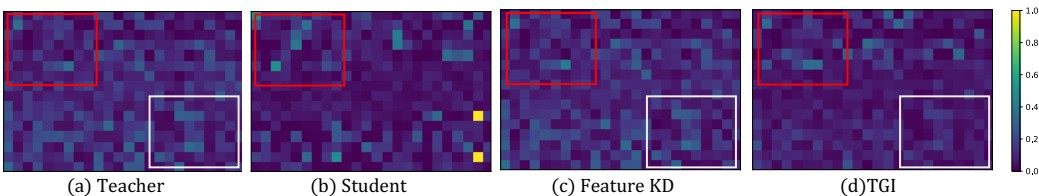

Figure 4: Visualization of channel-wise $L_1$ norm for distillation from CP-Pillar to CP-Pillar-v0.64.

**Teacher Guided Initialization.** As mentioned in Section 4.2, feature KD itself outperforms other KD methods, but its deficient synergy results hinder 3D detection KD pipeline from achieving promising performance. This might be caused by the conflict between different optimization directions on

intermediate features from five KD loss terms in Eq. (4). Hence, we explore whether there is a substitute manner which can also leverage teacher's guidance on the feature extraction aspect. Since the value of feature map is determined by model's weights, finding hints from teacher weights becomes an alternative to guide student's feature extraction.

Hence, we propose to directly use the trained weights of teacher to serve as the initialization of student network, named teacher guided initialization (**TGI**) to enhance student model's feature extraction abilities by inheriting it from a teacher model. Although such initialization is intuitive for input compressed students (*e.g.* CP-Pillar-v0.64), it cannot be directly applied to pruned students (*e.g.* CP-Voxel-S). Therefore, we employ the parameter remapping strategy FNA [8] to project teacher weights on student parameters. Take 2D convolution layer as an example, teacher and student parameters are represented with $W^t \in \mathbb{R}^{r \times q \times h \times w}$ and $W^s \in \mathbb{R}^{v \times u \times h \times w}$, where $v \leq r$ and $u \leq q$, respectively. The first $v$ and $u$ channels of teacher parameters are directly assigned to the slimmed student network in FNA. We visualize the channel-wise $L_1$ norm of backbone feature to compare TGI extracted features with other strategies. As shown in Figure 4, our TGI extracts feature similar to teacher's (see red box region), but not naively mimics all teacher channels as feature KD does (see white box region). We argue that fully imitating all teacher features might penalize student optimization regarding the architecture or input resolution discrepancy between teacher-student pairs. We empirically show that our simple TGI strategy achieves comparable or even better performance individually among four feature KD methods and shows prominent synergy effects on six teacher-student pairs (See Suppl.).

## 5.2 Main Results and Comparison

To verify our improved KD pipeline, we conduct experiments on all six teacher-student pairs. As shown in Table 3, our improved KD pipeline consistently surpasses previous KD strategies on all settings with $0.7\% \sim 1.9\%$ improvement, thanks to our enhanced logit KD and more collaborative TGI. Equipping our lightweight detectors with the improved KD pipeline, CP-Voxel-S obtains comparable performance to CP-Voxel in terms of mAPH with around $2.4\times$ fewer flops and $1.6\times$ fewer activations. Furthermore, with $3.1\times$ fewer flops, CP-Voxel-XS only suffers $0.76\%$ performance degradation. In addition, with around $1.6\times$ fewer flops and $1.5\times$ fewer activations, our distilled CP-Pillar-v0.4 even slightly outperforms CP-Pillar. CP-Pillar-v0.64 requires only $25\%$ flops and $29\%$ activations of teacher model, while achieving $55.82\%$ mAPH, only $3.27\%$ performance drop compared to CP-Pillar. These experimental results demonstrate that knowledge distillation is a promising technique to improve the performance of efficient 3D detectors.

## 5.3 Comparison with Other Detectors

To further demonstrate the efficiency and effectiveness of our designed detectors and KD pipeline, we also compare our distilled students with other detectors on the full WOD. As shown in Table 5, our CP-Voxel-S even outperforms its teacher CP-Voxel with around $2.0\times$ fewer parameters, $2.4\times$ fewer flops and $1.6\times$ activations. With similar latency and much fewer parameters, flops as well as activations, CP-Voxel-XS outperforms SECOND by $7.6\%$. Our CP-Pillar-v0.64, is $2.4\times$ faster than PointPillar on a GTX-1060 while achieves $1.8\%$ higher performance.

Table 5: Comparison with other detectors on full WOD. † indicates results re-implemented by us.

| Detector | Params (M) | Flops (G) | Acts (M) | Latency (ms) | LEVEL 2 mAPH |
|---|---|---|---|---|---|
| PointPillar† [21] | 4.8 | 255.0 | 233.5 | 129.1 | 57.05 |
| SECOND† [46] | 5.3 | 84.5 | 76.4 | 84.6 | 57.23 |
| CP-Pillar† [49] | 5.2 | 333.9 | 303.0 | 157.9 | 61.56 |
| CP-Voxel† [49] | 7.8 | 114.8 | 101.9 | 125.7 | 65.58 |
| PV-RCNN [33] | 13.1 | 117.7 | 399.4 | 623.2 | 63.33 |
| PV-RCNN++† [34] | 16.1 | 123.5 | 179.7 | 435.9 | **69.46** |
| CP-Voxel-S + Ours | 4.0 | 47.8 | 65.7 | 98.0 | **65.75** |
| CP-Voxel-XS + Ours | 2.8 | 36.9 | 58.4 | 88.1 | 64.83 |
| CP-Voxel-XXS + Ours | 1.0 | 12.0 | 33.1 | 70.4 | 60.93 |
| CP-Pillar-v0.4 + Ours | 5.2 | 212.9 | 197.7 | 103.4 | 61.60 |
| CP-Pillar-v0.48 + Ours | 5.2 | 149.4 | 142.3 | 81.9 | 60.95 |
| CP-Pillar-v0.64 + Ours | 5.2 | 85.1 | 88.0 | 54.5 | 58.89 |

## 5.4 Cross Stage Distillation

A prevailing strategy to improve state-of-the-art 3D object detectors is to adopt a object proposal refinement head for two stage detection [33, 7, 34]. However, despite performance improvements by $3.5\%$, PVRCNN++ requires

Table 6: Cross stage 3D detector distillation.

| Detector | Params (M) | Flops (G) | Acts (M) | Latency (ms) | Mem. (G) | LEVEL 2 mAPH |
|---|---|---|---|---|---|---|
| PV-RCNN++ | 16.1 | 123.5 | 179.7 | 435.9 | 4.2 | 67.80 |
| CP-Voxel | 7.8 | 114.8 | 101.9 | 125.7 | 2.8 | 64.29 |
| CP-Voxel + Ours | 7.8 | 114.8 | 101.9 | 125.7 | 2.8 | **65.27** |

around $2.2\times$ parameters, $1.8\times$ activations and $3.5\times$ latency compared to CP-Voxel (see Table 6). Such computation and parameter overheads hinder the real-world applications of state-of-the-art two-stage 3D detectors. Therefore, here we also investigate whether the knowledge of the two-stage detector can help the learning of single-stage detector. It is noteworthy that this is the first attempt at cross stage distillation in both 2D and 3D object detection. As shown in Table 6, leveraging hints

from pretrained PVRCNN++, our distilled CP-Voxel achieves around $1\%$ performance gains without any extra computation and parameter overheads during inference.

## 5.5 Generalization to More Scenarios

To demonstrate the generality of our compression and knowledge distillation manners, we provide experiments on other dataset, detector and compression manners. Besides, we construct experiment and discussion on extending our method to other task and more detectors in the Suppl..

**Generality on Other Dataset and Detector.** As shown in Table 7 and Table 8, both our compression conclusion in Sec. 3.3 and the improved KD method can generalize well to KITTI [9] dataset with a new anchor-based detector SECOND. Especially, the SECOND (a) surpasses teacher performance by around $0.5\%$ with $3.5\times$ fewer flops, showing the generality of our conclusion and methods.

Table 7: Model and input compression results on KITTI. Teacher models are marked in gray.

| Detector | | Architecture | | | Efficiency | | | | Moderate mAP@R40 | CPR |
|---|---|---|---|---|---|---|---|---|---|---|
| | | Width | | Voxel Size | Params | Flops | Acts | Latency | | |
| | | PFE | BFE | (m) | (M) | (G) | (M) | (ms) | | |
| SECOND | | 1.00 | 1.00 | 0.05 | 5.3 | 80.5 | 69.3 | 77.4 | 67.24 | - |
| | (a) | 0.75 | 0.50 | 0.05 | 1.6 | 23.0 | 38.0 | 51.8 | 65.62 | 0.69 |
| | (b) | 0.50 | 0.50 | 0.05 | 1.4 | 20.5 | 35.9 | 46.1 | 64.21 | 0.68 |
| | (c) | 0.50 | 1.00 | 0.05 | 4.6 | 72.4 | 65.2 | 70.6 | 65.70 | 0.50 |
| | (d) | 1.00 | 1.00 | 0.10 | 5.3 | 21.2 | 19.4 | 34.2 | 54.32 | 0.62 |

Table 8: Knowledge distillation results for 3D detection on KITTI. Performance are measured in moderate mAP over 40 recall positions. Best method is noted by **bold**.

| Detector | No Distill | Logit KD | | Feature KD | | | | Label KD | Ours | Flops (G) | Acts (M) |
|---|---|---|---|---|---|---|---|---|---|---|---|
| | | KD | GID-L | FitNet | Mimic | FG | GID-F | KD | | | |
| SECOND | 67.24 | - | - | - | - | - | - | - | - | 80.5 | 69.3 |
| SECOND (a) | 65.62 | 66.06 | 66.34 | 66.00 | 66.37 | 66.58 | 66.75 | 67.03 | **67.70** | 23.0 | 38.0 |

**Generality on Advance Compression Manner.** As shown in Table 9, since the coarser-resolution detector has more architecture-level redundancy with less input information, CP-Pillar (f) achieves higher CPR by combining the model and input compression strategies. Besides, after applying KD methods to CP-Pillar (f) as Table 10, it is still more efficient though fewer improvements from KD strategies with less redundancy. This demonstrates that our pipeline can substantially obtain a more efficient detector by harvesting the progress of advanced compression and distillation manners.

Table 9: Integrating different compression manners on Waymo. Teacher model is marked in gray.

| Detector | Architecture | | | | Efficiency | | | | LEVEL 2 mAPH | CPR |
|---|---|---|---|---|---|---|---|---|---|---|
| | Width | | | Voxel Size | Params | Flops | Acts | Latency | | |
| | PFE | BFE | Head | (m) | (M) | (G) | (M) | (ms) | | |
| CP-Pillar | 1.00 | 1.00 | 1.00 | 0.32 | 5.2 | 333.9 | 303.0 | 157.9 | 59.09 | - |
| CP-Pillar-v0.4 | 1.00 | 1.00 | 1.00 | 0.40 | 5.2 | 212.9 | 197.7 | 103.4 | 57.55 | 0.64 |
| CP-Pillar (e) | 1.00 | 0.875 | 0.875 | 0.32 | 4.0 | 260.1 | 267.7 | 134.7 | 58.53 | 0.54 |
| CP-Pillar (f) | 1.00 | 0.875 | 0.875 | 0.40 | 4.0 | 163.9 | 175.5 | 92.1 | 57.36 | 0.67 |

## 6 Ablation Studies

In this section, we conduct extensive ablation experiments to in-depth investigate the effectiveness of each component in our improved KD pipeline.

**Component Ablation.** Here, we investigate each component of our KD method and their synergy results. As shown in Table 11, based on CP-Voxel-XXS, both PP logit KD and TGI can obtain around $1.4\%$ improvements separately. When incorporating with each other and label KD, they further obtain around $1.6\%$ gains, showing the prominent synergy impact of our components. On the contrary, feature KD even suffers $0.2 \sim 0.6\%$ performance drop when combined with other KD techniques.

**Investigation of TGI.** We study how different parameter remapping manners and teachers influence the TGI. Besides FNA, we also attempt other parameter remapping strategies for TGI, including OFA [3] and Slim [26] by selecting important channels with designed indicators. As shown in Table 12, based on CP-Voxel-XS, OFA and Slim obtain inferior results compared to FNA, since indicator guided channel selection cannot determine consistent channel mapping for skip connection [13], while FNA survives by simply selecting beginning channels of all layers. In addition, CP-Voxel-XXS

Table 10: Knowledge distillation results for more sophisticated compressed 3D detector on Waymo. Teacher model is marked by gray Best method is noted by **bold**. $\S$ indicates the CPR is calculated according to the performance of best distilled student.

| Detector | No Distill | Logit KD | | Feature KD | | | | Label KD | Ours | Flops (G) | Acts (M) | CPR$^\S$ |
|---|---|---|---|---|---|---|---|---|---|---|---|---|
| | | KD | GID-L | FitNet | Mimic | FG | GID-F | KD | | | | |
| CP-Pillar | 59.09 | - | - | - | - | - | - | - | - | 333.9 | 303.0 | - |
| CP-Pillar-v0.4 | 57.55 | 57.51 | 57.54 | 57.89 | 58.57 | 58.44 | 58.26 | 58.10 | **59.24** | 212.9 | 197.7 | 0.68 |
| CP-Pillar (f) | 57.36 | 56.93 | 56.70 | 57.15 | 57.81 | 57.48 | 57.77 | 57.57 | **58.62** | 163.9 | 175.5 | 0.70 |

Table 11: Component ablation study based on CP-Voxel-XXS.

| PP logit KD | Label KD | TGI | Feature KD | mAPH |
|---|---|---|---|---|
| | | | | 56.26 |
| √ | | | | 57.68 |
| | | √ | | 57.61 |
| √ | | √ | | 58.83 |
| √ | √ | | | 58.49 |
| √ | √ | √ | | **59.28** |
| √ | √ | | √ | 58.28 |
| √ | √ | √ | √ | 58.67 |

Table 12: Different remap manners studies of TGI based on CP-Voxel-XS and CP-Voxel-XXS.

| Student | Teacher | Remap | mAPH |
|---|---|---|---|
| CP-Voxel-XS | - | - | 61.16 |
| | CP-Voxel | FNA | 62.43 |
| | CP-Voxel | OFA | 60.06 |
| | CP-Voxel | Slim | 61.74 |
| CP-Voxel-XXS | - | - | 56.26 |
| | CP-Voxel | FNA | 54.92 |
| | CP-Voxel-XS | FNA | 57.61 |

only improves by inheriting parameters from CP-Voxel-XS but not CP-Voxel, which indicates a large architecture discrepancy between teacher and student hinders the effectiveness of TGI.

**Different Variants of PP Logit KD.** Here, we analyze different variants of PP logit KD (*i.e.* pivotal position selection according to teacher confidence, response ranking or soft Gaussian instance mask) in Section 5.1 based on CP-Voxel-S. While previous coarse instance-aware response imitation method GID-L obtains around $0.7\%$ improvements, all three PP logit KD variants achieve around $1.6\% \sim 1.9\%$ gains. This demonstrates the significance of focusing on only vital positions in output-level distillation for 3D detection.

Table 13: Analysis of different PP logit KD variants based on CP-Voxel-S.

| Logit KD Manner | mAPH |
|---|---|
| None | 62.23 |
| GID-L [6] | 62.89 |
| Confidence PP | 63.84 |
| Rank PP | 63.85 |
| Gaussian PP | **64.16** |

Table 14: Analysis of label KD based on CP-Pillar-v0.48.

| | Ground Truth | | Teacher Pred | | mAPH |
|---|---|---|---|---|---|
| | Cls | Reg | Cls | Reg | |
| (a) | all | all | no | no | 56.24 |
| (b) | all | all | all | no | 56.47 |
| (c) | all | non-overlap | all | all | 56.99 |
| (d) | all | all | all | non-duplicate | 56.74 |
| (e) | all | all | all | all | **57.54** |

**Analysis of Label KD.** Although label KD has been proposed by the recent work [28], it just attributed its improvement to the "dark knowledge" from teacher. Here, we attempt to analyze how teacher predictions help student based on CP-Pillar-v0.48, where label KD performs best. As shown in Table 14, only integrating teacher prediction for classification label assignment, (b) achieves minor gains, which indicates that label KD mainly benefits the regression objective of student. When removing GTs highly overlapped with teacher predictions, (c) still achieves around $0.8\%$ gains, which demonstrates that more achievable regression targets from teacher prediction facilitate student optimization. Last, by removing teacher predictions with object centers in the same BEV grid (*i.e.* duplicate objectives for single position), we find that (d) suffers performance drop compared to (e), which shows that duplicate regression objectives can also boost student regression ability.

## 7 Conclusion

We have examined the potential of knowledge distillation to serve as a generic method for obtaining efficient 3D detectors with extensive experimental results and analysis. We found that pillar-based detector prefers input compression while voxel-based detector is more suitable for width compression in designing efficient student models. Besides, we proposed pivotal position logit KD and teacher guided initialization for enhancing the 3D KD pipeline. Our best performing detector outperforms its teacher with $2.4\times$ fewer flops and our most efficient detector is $2.2\times$ faster than previous fastest voxel/pillar-based detector PointPillars on an NVIDIA A100 with higher performance. We hope our benchmark and analysis could inspire future investigations on this problem.

**Acknowledgement.** This work has been supported by Hong Kong Research Grant Council - Early Career Scheme (Grant No. 27209621), HKU Startup Fund, and HKU Seed Fund for Basic Research.

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
