# Supplementary Material for
# Towards Efficient 3D Object Detection with Knowledge Distillation

**Jihan Yang[1]**    **Shaoshuai Shi[2]**    **Runyu Ding[1]**    **Zhe Wang[3]**    **Xiaojuan Qi[1]**

[1]The University of Hong Kong    [2]Max Planck Institute for Informatics    [3]SenseTime Research

{jhyang, ryding, xjqi}@eee.hku.hk, {shaoshuaics, wzlewis16}@gmail.com

## Outline

In this supplementary file, we provide more details and experiments not elaborated in our main paper due to page length limits:

- Sec. S1: Implementation details and hyper-parameters of our 3D detection KD benchmark.
- Sec. S2: Additional experimental results on synergy results of TGI, per-class results, error bar results, and focal loss attempts.
- Sec. S3: More analysis, including latency comparisons on different accelerators, operation-level optimizations and detectors, and qualitative analysis of CPR.
- Sec. S4: Generality of our method on 3D semantic segmentation.
- Sec. S5: Discussion on other detectors such as sparse detection architectures and other input representations.
- Sec. S6: Limitation analysis.

## S1    Implementation Details for Our Benchmark

In this section, we describe the implementation of previous 2D KD methods in 3D object detection. Notice that most 2D detection KD methods are built on anchor-based detectors (*e.g.* Faster-RCNN [17]) and model compressed teacher-student pairs, so we modify them to adapt to anchor-free detectors and handle input resolution compression. On the other hand, we also provide detailed hyper-parameter values to help reproduce our results. Besides, we will also open-source our benchmark suit upon acceptance. Most of the experiments are trained with 8 NVIDIA 1080Ti, while a few experiments are trained with 8 NVIDIA V100 or 8 NVIDIA A100. Full set results on Waymo are trained with 16 NVIDIA 1080Ti or V100.

**Logit KD.** As for logit KD methods (*i.e.* vanilla KD [9] and GID-L [6]), vanilla KD use all ones $m_{cls}$ to fully mimic all teacher outputs and set the mask $m_{cls}$ to be all one. As for GID-L, the original anchor-wise region selection manner cannot be extended to input resolution compressed students, since the interpolation cannot handle the resolution mismatch of regression predictions $p_{reg}$ from teacher and student models. In this regard, we refer to the ablation studies of the original paper and use ground truth boxes as critical region selection criteria. Specifically, we set the non-zero spatial positions in the assigned classification target heatmap to one in $m_{cls}$. The loss weight $\alpha_1$ and $\alpha_2$ of $\mathcal{L}_{cls}^{KD}$ and $\mathcal{L}_{reg}^{KD}$ are set to 15.0 and 0.2, respectively. Notice that the loss weight $\alpha_2$ of regression term $\mathcal{L}_{reg}^{KD}$ in Eq. (4) will be set to 0 for input resolution compressed students.

As for our PP logit KD, the threshold of confidence PP is set to 0.3 by default and the rank $K$ for rank PP is set to 500. Although the three variants of PP logit KD show similar performance, we use Gaussian PP by default since it does not need hyper-parameters adjustment when adopted by different teacher-student pairs.

36th Conference on Neural Information Processing Systems (NeurIPS 2022).

**Feature KD.** For the implementation of different feature KD methods with Eq. (3), we only employ convolutional block $\phi$ for width compressed students to align the number of channels between $f^s$ and $f^t$ and only hire spatial interpolation $\kappa$ for input compressed students. Besides, for methods that utilize RoI Align $\psi$ to extract object-level features, we will not use the interpolation $\kappa$ to avoid introducing extra interpolation errors. As for the implementation of each method, we directly align student and teacher full features without mask $m_{\text{feat}}$ and RoI Align $\psi$ in FitNet [18]. As for Mimic [11], we use the more sophisticated RoI Align $\psi$ instead of the original spatial pyramid pooling to extract features for each GT, and construct imitation on the object-level features between teacher and student. As for FG [23], to extend its anchor-based critical region selection, we directly set the non-zero regions in the assigned classification target heatmap as the critical regions in $m_{\text{feat}}$, the other implementations are the same as FitNet. As for GID-F [6], we use teacher predictions after Non-Maximum Suppression (NMS) as critical regions and set their corresponding spatial positions in $m_{\text{feat}}$ to one. Besides, we utilize RoI Align $\psi$ to extract object-level features to calculate feature distillation loss and also apply the relation loss among different object features as the description in the original paper. The loss weight $\alpha_3$ of feature KD loss $\mathcal{L}_{\text{feat}}^{\text{KD}}$ is set to $100$ for input compressed students or $200$ for width compressed students, respectively. The loss weight of the relation loss of GID-F is set to $0.1$.

**Label KD.** There is only one work for label KD which has been described in the main paper. Notice that we do not hire NMS for teacher predictions for label assignment empirically. The score threshold $\tau$ to filter high-quality teacher predictions is set to $0.6$ by default.

Among different 2D KD methods, we notice that FG [23], Mimic [11] and GID [6] highlight the critical region selection on feature KD or logit KD to tackle the imbalance between foreground and background regions in object detection. There are mainly two foreground-region imitation strategies: one is using RoI Align $\psi$ to extract object-wise features with teacher prediction or GT boxes as guidance (*e.g.* Mimic [11] and GID-F [6]); the other is assigning a one-hot mask $m_{\text{feat}}$ or $m_{\text{cls}}$ to calculate imitation loss on some critical regions (*e.g.* FG [23] and GID-L [6]). We empirically find that RoI Align is more suitable for input compression setups since it avoids the interpolation errors when aligning spatial resolutions while the mask-based strategy is more flexible and general as it allows position-wise imitation. Comparing different critical region selection techniques, we empirically show that a key factor to make KD methods work well on 3D detection is to focus on only a few positions. For example, our proposed PP logit KD focus on only around $\frac{1}{5} \sim \frac{1}{20}$ positions of the positions selected by traditional 2D KD methods. This is caused by the fact that a spatial position in the 3D BEV features can represent a $0.8m \times 0.8m \times 6m$ pillar in the 3D geometry space, which is informative and can cover even a single pedestrian. In this regard, the critical region selection techniques are supposed to focus on fewer informative positions in the 3D detection setting.

## S2 Additional Experimental Results

In this section, we provide some additional experimental results as a supplement to our main paper. This part consists of the full synergy results of TGI on six teacher-student pairs, per-class performance and error bar results.

### S2.1 Synergy Results of TGI

Table S1: Synergy results of TGI and feature KD on Waymo with six teacher-student pairs. Performance are measured in LEVEL 2 mAPH. Teacher results are masked by gray.

| Detector | No Distill | Feature KD | Label KD | PP Logit KD | TGI | PP Logit KD + Feature KD | PP Logit KD + TGI | Label KD + Feature KD | Label KD + TGI |
|---|---|---|---|---|---|---|---|---|---|
| CP-Pillar | 59.09 | - | - | - | - | - | - | - | - |
| CP-Pillar-v0.4 | 57.55 | 58.57 | 58.10 | 58.21 | 59.03 | 58.18 | 59.24 | 58.35 | 59.19 |
| CP-Pillar-v0.48 | 56.27 | 57.26 | 57.54 | 56.89 | 57.91 | 57.11 | 58.20 | 57.43 | 58.34 |
| CP-Pillar-v0.64 | 52.81 | 53.83 | 53.78 | 54.32 | 54.30 | 54.14 | 55.55 | 54.24 | 55.59 |
| CP-Voxel | 64.29 | - | - | - | - | - | - | - | - |
| CP-Voxel-S | 62.23 | 63.35 | 63.31 | 64.16 | 63.48 | 63.58 | 64.18 | 62.62 | 63.50 |
| CP-Voxel-XS | 61.16 | 62.48 | 61.81 | 62.76 | 62.43 | 62.90 | 63.41 | 62.34 | 62.85 |
| CP-Voxel-XXS | 56.26 | 57.92 | 57.02 | 57.68 | 57.61 | 58.19 | 58.83 | 57.06 | 57.71 |

The poor synergy effect of feature KD is the main motivation for us to design TGI. Due to the page limitation, we only present its experimental results on CP-Voxel-XXS as an example. Here, we

compare the synergy results of feature KD and TGI on six teacher-student pairs to further show the promising performance of our TGI. As shown in Table S1, the results obtained by combining TGI and label KD or PP logit KD consistently outperform the synergy results of feature KD on all 12 scenarios, manifesting that TGI collaborates better with other KD techniques. Furthermore, our TGI itself achieves comparable results or even surpasses feature KD among six teacher-student pairs. These experimental results strongly demonstrate that our proposed TGI can be a powerful substitute for feature KD to transfer the feature extraction ability from the teacher model.

## S2.2  Per-class Performance

Table S2: Per-class performance on full Waymo dataset for our six distilled efficient student models. Performance are measured in mAP/mAPH. Teacher results are masked by gray. Best results are indicated by **bold**.

| Detector | Vehicle | | Pedestrian | | Cyclist | |
|---|---|---|---|---|---|---|
| | LEVEL 1 | LEVEL 2 | LEVEL 1 | LEVEL 2 | LEVEL 1 | LEVEL 2 |
| CP-Pillar | 72.75/72.24 | 64.48/64.02 | 74.01/64.06 | 65.74/56.76 | **67.84/66.37** | **65.34/63.92** |
| CP-Pillar-v0.4 + Ours | **73.01/72.46** | **64.85/64.36** | **75.00/64.24** | **66.86/57.13** | 67.18/65.76 | 64.69/63.32 |
| CP-Pillar-v0.48 + Ours | 72.42/71.85 | 64.42/63.89 | 74.38/63.74 | 66.26/56.62 | 66.19/64.76 | 63.72/62.34 |
| CP-Pillar-v0.64 + Ours | 71.37/70.77 | 63.30/62.75 | 71.45/61.05 | 63.22/53.86 | 63.87/62.39 | 61.48/60.05 |
| CP-Voxel | **74.31/73.75** | **66.35/65.84** | 76.19/70.10 | 68.44/62.82 | 71.76/70.63 | 69.16/68.07 |
| CP-Voxel-S + Ours | 74.28/73.72 | 66.17/65.66 | **76.72/70.68** | **68.96/63.37** | **71.97/70.81** | **69.36/68.24** |
| CP-Voxel-XS + Ours | 73.62/73.05 | 65.53/65.01 | 75.50/69.29 | 67.67/61.96 | 71.30/70.09 | 68.69/67.52 |
| CP-Voxel-XXS + Ours | 69.20/68.55 | 61.15/60.57 | 71.76/64.95 | 63.71/57.53 | 68.51/67.18 | 65.98/64.70 |

We report the per-category performance of our efficient detectors on full Waymo Open Dataset [21] in Table S2. As illustrated in Table S2, comparing CP-Pillar-v0.64 and CP-Pillar, the performance gap mainly lies in pedestrians and cyclists (around 3% gap), while vehicle suffers around 1.5% gap. This might be caused by the fact that coarser input resolution penalizes the performance of small objects such as pedestrians and cyclists more severely. As for voxel-based detector CP-Voxel-XXS, we notice that its performance gap from teacher distributes more evenly on different categories than CP-Pillar-v0.64, as the model width compression does not have special penalization on any categories.

## S2.3  Error Bar

Table S3: Repeat results of our different models on Waymo . We report the reproduced results with 5 rounds as well as their averaged results and standard variance. The performance is measured in LEVEL 2 mAPH.

| Detector | Round 1 | Round 2 | Round 3 | Round 4 | Round 5 | Average | Standard Variance |
|---|---|---|---|---|---|---|---|
| CP-Pillar | 59.09 | 59.13 | 59.13 | 59.14 | 59.01 | 59.10 | 0.05 |
| CP-Pillar-v0.64 | 52.81 | 52.75 | 52.85 | 52.85 | 53.25 | 52.90 | 0.20 |
| CP-Pillar-v0.64 + Ours | 55.75 | 55.82 | 56.02 | 55.75 | 55.73 | 55.81 | 0.12 |

Here, to show the robustness of our experimental results, we reproduce knowledge distillation on CP-Pillar-v0.64 five times and report the average and standard derivation of performance. As shown in Table S3, the performance of our distilled CP-Pillar-v0.64 is more stable than the student without distillation, which indicates that our improved KD pipeline can boost performance stably.

## S2.4  Focal Loss Results

As focal loss [13] is a widely-used solution for the foreground and background region imbalance issue, it is intuitive to also employ it as the distillation loss. In this regard, here we provide an experimental comparison between focal loss and our proposed PP logit KD for logit KD. As shown in Table S4, PP logit KD is around 0.7% and 8.2% higher than focal loss on CP-Voxel-XS and CP-Pillar-v0.64, respectively. As for CP-Pillar-v0.64, since the capability difference between teacher and student are large, focal loss even suffers performance degradation compared to vanilla KD, while our PP logit KD consistently brings performance boost. The reason for the inferior performance of focal loss for distillation is that it will emphasize regions that are most different among teacher and student pairs but not most information-rich areas. Those large prediction difference areas could be caused by

the capability gap between teacher and student and thus renders focal loss a suboptimal strategy for student learning.

Table S4: Results of leveraging focal loss as logit distillation loss on Waymo. Teacher models are marked in gray.

| Detector | No Distill | KD [9] | Focal loss | PP Logit KD |
|---|---|---|---|---|
| CP-Voxel | 64.29 | - | - | - |
| CP-Voxel-XS | 62.23 | 62.81 | 63.48 | 64.16 |
| CP-Pillar | 59.09 | - | - | - |
| CP-Pillar-v0.64 | 52.81 | 50.78 | 46.11 | 54.32 |

## S3    More Analysis

In this section, we provide some investigations on the influence of accelerator types and operation-level optimizations on the measured latency as well as the qualitative analysis of our proposed CPR.

### S3.1    Latency Analysis

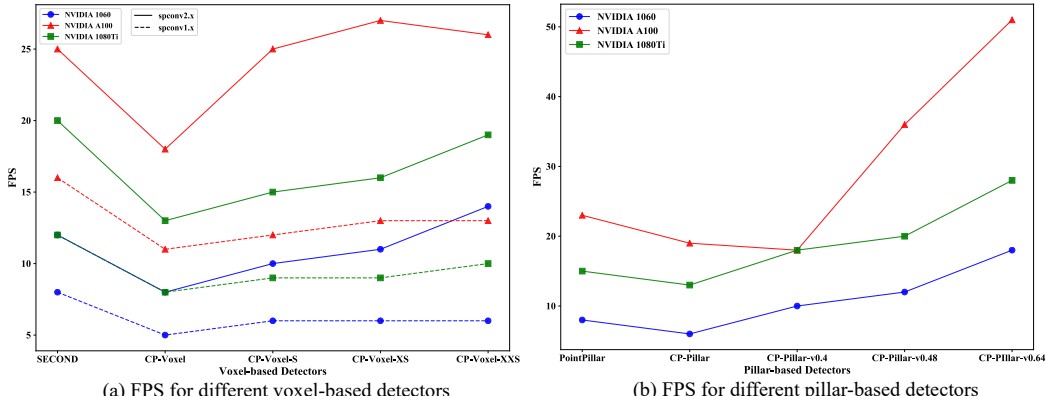

(a) FPS for different voxel-based detectors      (b) FPS for different pillar-based detectors

Figure S1: Comparison on the FPS with different hardware devices and operation-level optimizations for different detectors.

Table S5: Information of the inference machine.

| Type | GPU | CPU |
|---|---|---|
| Personal Computer | NVIDIA GTX-1060 | Intel(R) Core(TM) i7-7700K CPU @ 4.20GHz |
| Server#1 | NVIDIA GTX-1080Ti | Intel(R) Xeon(R) CPU E5-2682 v4 @ 2.50GHz |
| Server#2 | NVIDIA A100 | AMD EPYC 7742 64-Core |

Inference time, latency or FPS directly measure the execution speed of a model on a given hardware configuration, and have been widely adopted to assess the model efficiency in 3D detection. However, different papers [28, 27, 7] measure the latency based on different machines, hindering the fair comparisons and standardization among different approaches. Besides, we empirically show that the operation-level optimization has a large impact on the latency measurement and even influences the conclusion. In this regard, we investigate how different hardware devices and operation-level optimizations (*i.e.* sparse convolution [8]) affect the latency measurement in terms of FPS. The basic information of our tested machines is shown in Table S5, including one personal computer and two servers.

**Influence of Hardware Devices.** As shown in Figure S1 (a) and (b), detectors run faster on more powerful GPU consistently for both voxel-based and pillar-based detectors, which demonstrate the great influence of hardware devices. In addition, we notice that some compressed detectors meet negative optimization (*i.e.* model needing fewer computations has larger latency) on the latest GPU architecture NVIDIA-A100 (see CP-Voxel-XS *vs.* CP-Voxel-XXS and CP-Pillar *vs.* CP-Pillar-v0.4), which has not been observed on GTX-1060 and GTX-1080Ti. This might be caused by the different underlying implementation strategies at the hardware level. This also verifies that latency-orientated efficient detector designs largely depend on the type of hardware devices.

**Operation-level Optimization.** Sparse convolution network [8] is a major component in existing voxel-based detectors to efficiently extract voxel-wise features from voxelized point clouds. As they are not fully-optimized toward hardware, it occupies a large percentage of latency for voxel-based detectors using A popular implementation Spconv [1], although the GLOPs are not that high. Here, we also investigate how different implementations of the sparse convolution influence the measured latency. As shown in Figure S1 (a), voxel-based detectors implemented with Spconv2.x is much faster than their counterparts with Spconv1.x (Spconv2.x is the optimized version of Spconv1.x). Moreover, the operation-level optimization can even have a larger impact than the hardware devices (see NVIDIA 1080Ti with Spconv1.x and NVIDIA 1060 with Spconv2.x). In addition, although the width-level compressed students of CP-Voxel require significantly fewer flops, parameters and activations compared to CP-Voxel, they cannot obtain obvious speed up on the latency with Spconv1.x, which indicates that non-parametric computations (*i.e.* computation not directly related to learnable parameters) occupy most of the latency for voxel-based detectors using Spconv1.x. As a consequence, based on NVIDIA 1060, CP-Voxel with Spconv2.x runs faster than CP-Pillar, while runs slower than CP-Pillar when using Spconv 1.x. This demonstrates that testing model latency on different operation-level optimizations can draw totally different conclusions when comparing the efficiency of different detectors in terms of latency.

Besides the above two factors, we also observe that even hardware status (*e.g.* temperature) could influence the final obtained latency, indicating that the latency is hard to stably reproduce on the same machine and cannot serve as a standard measurement on model efficiency. In this regard, we focus on the parametric measurement such as flops and activations in the main paper, as they will not be influenced by the above hardware, software or environment level factors.

### S3.2 Qualitative Analysis of CPR

Table S6: More model compression results. Teacher models are marked in gray. See text for details.

| Detector | | Architecture | | | | | Efficiency | | | | | LEVEL 2 mAPH | CPR |
|----------|---|-----|-----|------|-----|-----|----------|-------|------|---------|------|------|-----|
| | | Width | | | Depth | | Params | Flops | Acts | Latency | Mem. | | |
| | | PFE | BFE | Head | PFE | BFE | (M) | (G) | (M) | (ms) | (G) | | |
| CP-Pillar | | 1.00 | 1.00 | 1.00 | 1.00 | 1.00 | 5.2 | 333.9 | 303.0 | 157.9 | 5.2 | 59.09 | - |
| | (a) | 1.00 | 0.50 | 1.00 | 1.00 | 1.00 | 1.5 | 130.1 | 203.1 | 97.1 | 3.6 | 55.35 | 0.58 |
| | (b) | 1.00 | 0.25 | 0.25 | 1.00 | 1.00 | 0.3 | 23.8 | 91.2 | 51.9 | 2.3 | 46.16 | 0.59 |
| | (c) | 1.00 | 1.00 | 1.00 | 1.00 | 0.50 | 2.2 | 258.5 | 234.1 | 118.5 | 4.3 | 55.24 | 0.52 |
| | (d) | 1.00 | 1.00 | 1.00 | 1.00 | 0.33 | 1.4 | 234.6 | 210.0 | 107.8 | 4.0 | 47.97 | 0.42 |
| CP-Voxel | | 1.00 | 1.00 | 1.00 | 1.00 | 1.00 | 7.8 | 114.7 | 101.9 | 125.7 | 2.8 | 64.29 | - |
| | (a) | 0.50 | 0.50 | 0.50 | 1.00 | 1.00 | 1.9 | 28.8 | 51.2 | 75.1 | 1.7 | 59.47 | 0.64 |
| | (b) | 0.50 | 0.25 | 0.25 | 1.00 | 1.00 | 1.0 | 12.0 | 33.1 | 70.4 | 1.3 | 56.26 | 0.67 |
| | (c) | 1.00 | 1.00 | 1.00 | 0.50 | 0.50 | 3.0 | 63.9 | 65.2 | 73.0 | 1.9 | 60.95 | 0.61 |
| | (d) | 1.00 | 1.00 | 1.00 | 0.33 | 0.33 | 1.8 | 47.9 | 52.2 | 59.0 | 1.6 | 55.78 | 0.57 |

When designing efficient student models, we propose CPR to quantitatively measure the trade offs between efficiency and performance of a compressed student model. Here, we take some examples to qualitatively analyze the correlation between the CPR, the model efficiency as well as the model accuracy. As shown in Table S6, comparing CP-Pillar (a) and (c), they achieve similar performance, but CP-Pillar (a) requires only half of flops and fewer parameters, activations, latency and training memory. This indicates that CP-Pillar (a) achieves better trade offs between accuracy and efficiency, which can be reflected on its higher CPR. Similar conclusions can be drawn by comparing CP-Pillar (b) and (d), CP-Voxel (a) and (c), as well as CP-Voxel (b) and (d). These qualitative results demonstrate the good correlation between CPR and the compromise between accuracy and efficiency for a given compressed model.

## S4 Generality on 3D Semantic Segmentation

In this work, the above experiments are all built on 3D object detection, which is a sparse prediction task. However, we argue that our sparse distillation manner (*i.e.* pivotal position logit KD) can also generalize to dense prediction tasks such as 3D semantic segmentation. As the student model has dense GTs supervision in training, dense distillation loss on massive uninformative points and regions, such as road points, might be redundant and can overwhelm the overall distillation loss. Instead, our

---

[1]https://github.com/traveller59/spconv

sparse distillation might help the student focus on more important areas by using teacher prediction as regularization.

Here, we follow the design principle of PP logit KD and adapt it to handle the dense semantic segmentation task. We apply distillation loss on points with predictions that are correct but less confident than the teacher. Our simple design is motivated by three intuitions: ($i$) Points that are correctly predicted with lower confidence are often some challenging cases that the model is struggling but also has the capability to handle. By harvesting knowledge from a high-performing teacher model, the student can learn to match the confidence level of the teacher which provides more information than the one-hot GT. ($ii$) Points that are correctly predicted with higher confidence are often easy samples that have very close prediction confidence to the teacher model. Considering that these samples are already handled well by the model, they have low chance to benefit from distillation but might cause redundancies. ($iii$) Points that are incorrectly predicted by the student are often cases that might be out of the ability of student models. Specifically, we have the confidence of student predictions $conf^s$, the confidence of teacher predictions $conf^t$ and a pre-defined threshold $\tau$. We will only apply distillation loss for student predictions that are correct and have $conf^s + \tau < conf^t$.

We also provide experimental results for our design on the 3D semantic segmentation dataset ScanNet [5]. Here, we use a small version of MinkowskiNet [4] for fast verification. On the one hand, as shown in Table S7, we try both model width and input resolution compression to obtain efficient student models, and select MinkowskiNet14-v0.04 as the student model for KD due to its higher CPR. On the other hand, as shown in Table S8, we compare the effectiveness of KD [9], PP logit KD and TGI on MinkowskiNet14-v0.04, where both our proposed PP logit KD and TGI obtain improvements. In particular, our sparse PP logit KD surpasses the dense logit KD method with around 0.8% gains. Our statistics also show that our PP logit KD only leverages 19.03% points for distillation at the first epoch and 3.66% points for distillation at the last epoch. These experiments and statistics demonstrate that sparse distillation can also work on the dense prediction task.

Table S7: Model width and input resolution compression results of MinkowskiNet14 on ScanNet. The teacher model is marked in gray.

| Architecture | | | Efficiency | | | mIoU | CPR |
|---|---|---|---|---|---|---|---|
| Model | Width | Voxel Size (m) | Params (M) | Flops (T) | Acts (M) | | |
| MinkowskiNet14 | 1.0 | 0.02 | 1.7 | 46.2 | 27.9 | 65.77 | - |
| MinkowskiNet14-w0.5 | 0.5 | 0.02 | 0.5 | 18.2 | 17.4 | 61.84 | 0.60 |
| MinkowskiNet14-v0.04 | 1.0 | 0.04 | 1.7 | 5.7 | 8.9 | 62.82 | 0.78 |

Table S8: Knowledge distillation results of compressed MinkowskiNet14 on ScanNet.

| Model | Role | No Distill | KD [9] | PP Logit KD | TGI | Flops (T) | Acts (M) |
|---|---|---|---|---|---|---|---|
| MinkowskiNet14 | Teacher | 65.77 | - | - | - | 46.2 | 27.9 |
| MinkowskiNet14-v0.04 | Student | 62.82 | 63.65 | 64.40 | 64.22 | 5.7 | 8.9 |

## S5   Discussion on Other Detectors

In this work, we mainly focus on dense detectors (*e.g.* CenterPoint [27], PV-RCNN++ [19], SEC-OND [25]) with the most popular pillar/voxel input representations. Here, we construct some discussion for the knowledge distillation on the sparse detectors and other input representations to further demonstrate the generality of our distillation manners.

### S5.1   Discussion for Sparse Detectors

As the emergency of DETR [2], object detectors that directly produce sparse prediction without post-processing become a new popular detection paradigm. Although we only try our KD manner on dense detector in the main paper, we argue that our sparse distillation is still applicable for sparse transformer-based detector such as DETR [2], Deformable DETR [29], Object DGCNN [24], etc.

On the one hand, sparse transformer-based detectors that directly make instance predictions actually rely on learning to some sparser reference points and corresponding position features. For example, each object query in Deformable DETR [29] or Object DGCNN [24] is decoded into a reference point and neighboring points in order to focus only on those most informative positions. On the other hand,

although sparse detectors can directly generate sparse instance predictions, our sparse distillation (*i.e.* pivotal position KD) focuses on sparser and more fine-grained position-level information (see Figure S2). In this regard, it should still be applicable to sparse models with some specific modifications.

Here, we take Object DGCNN [24] as an example and provide two possible sparse distillation designs.

(1) As the transformer encoder and decoder of Object DGCNN are similar to Deformable DETR, it can be simply extended to a two-stage variant as Deformable DETR. In the two-stage variant, the transformer encoder will regard each pixel as an object query and construct a dense scoring on it, where top-score positions are picked as reference points. This is similar to our designed rank PP KD which enforces the student to imitate the prediction of teacher top-rank positions. Therefore, we can directly apply our sparse rank PP KD to those dense scoring predictions between teacher and student. Besides, we will also carry on feature imitation on those teacher top-ranked positions between teacher and student.

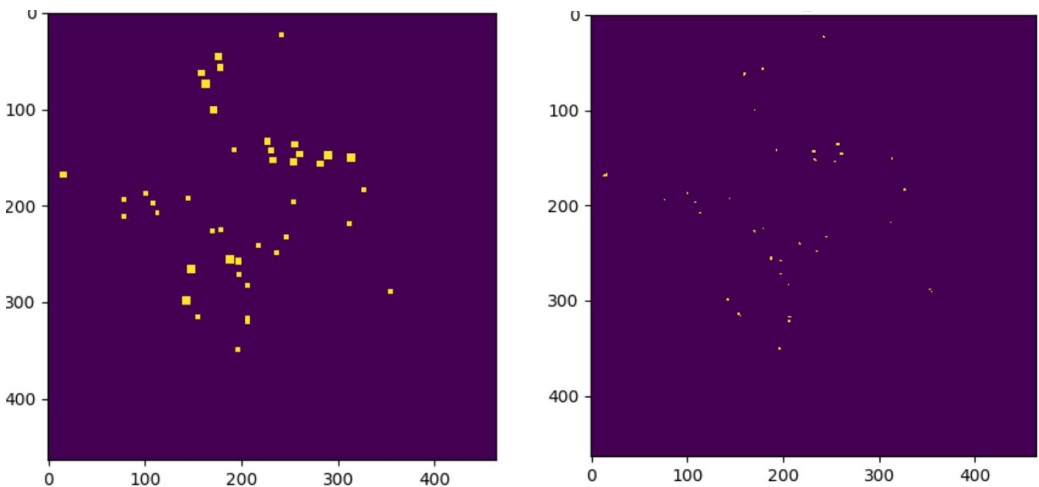

Figure S2: Visualization comparison of the imitation position (yellow positions) between instance-wise KD and our PP KD on the bird's eye view. Left: valid imitation regions for instance-wise KD. Right: valid imitation positions for our PP KD. Our PP KD has more fine-grained imitation regions compared to instance-wise KD. Best viewed in color.

(2) As for the one-stage variant of sparse detectors, learnable object queries will be decoded into reference points and neighboring points, so the sparse distillation can be constructed on those points and their corresponding BEV features. Specifically, we can first match the positive object queries of teacher and student as query pairs by checking whether they are matched to the same GT box. Then, we can enforce the decoded reference and neighboring points of the student to mimic their paired teacher counterparts. Besides, we will construct imitation on BEV features of those reference and neighboring positions between teacher and student.

### S5.2 Discussion for other input representations

Apart from the most popular pillar/voxel based object detectors discussed in the main paper, there are also point-based and range image based detectors. Therefore, we also provide the discussion on the point-based and range-based detectors here. As the TGI and label KD are detector-agnostic distillation manners and can be easily extended to detectors with any input representations, we only discuss the sparse distillation – pivotal position KD here.

Point-based detector [20, 3, 26, 28] take raw point clouds as input and and employed PointNet++ [16] to extract point features and generate point-wise object proposals. Range image based detectors leverage the native and dense representation for 3D points captured from LiDAR [15, 12, 1, 22]. As for the knowledge distillation on point-based and range image based detectors, since they still need to generate dense point-wise object proposals, our sparse distillation can still directly apply to it by selecting confident or top-ranked teacher positions for imitation (*i.e.* Confidence PP and Rank PP

in Table **??**). In this regard, our distillation strategies should be generalizable to all existing input representations.

## S6    Limitations

Although our work has already investigated the compression on model width, model depth and input resolution for designing lightweight student detectors, there are also exhausted layer-wise model compression methods [14, 10], which have not been attempted in this work. Although missing the compression attempts in such perspective, we argue that our paper mainly focuses on exploring the potential of knowledge distillation to obtain efficient 3D detectors and the existing compression attempts already fulfill our demands. Besides, we believe that the further progress and investigation of designing efficient 3D detectors is orthogonal to our KD attempts and can cooperate with our improved KD pipeline to obtain more efficient detectors.