# OpenReview forum: "Towards Efficient 3D Object Detection with Knowledge Distillation"
_NeurIPS.cc/2022/Conference — NeurIPS 2022 Accept_

### Official Review · Reviewer_nWAU · 2022-07-11

**Rating:** 8
**Confidence:** 2
**Soundness:** 4 excellent
**Presentation:** 4 excellent
**Contribution:** 3 good

**Summary:**

This work comprehensively looks into the state of knowledge distillation as it applies to point cloud 3D object detection. In particular, the authors argue for knowledge distillation’s use case for model compression. They begin by analyzing how different compression techniques, both model- and input-level affect performance by looking into two CenterPoint based detectors. Next, they benchmark a strong suite of current knowledge distillation methods. The authors also propose their own KD method combining the strengths of prior KD paradigms (logit KD, label KD, and teacher-guided initialization), showing superior performance over the baselines on the Waymo dataset.

**Questions:**

In Section 5.3, it is observed that the student model can outperform the teacher model. Is there intuition for why this may be? Intuitively, unless the student model has access to labels, it should not have enough information to correct these mistakes.

**Limitations:**

The authors do not explicitly address the limitations in their work.

**Strengths And Weaknesses:**

The paper is very well organized and clearly written, and each component of the research question seems thoroughly analyzed. Reading through the paper was a self-containing opportunity to learn about the field of KD in 3D object detection. The analysis section is thorough; the work not only benchmarks the different baselines, they also organize them according to the KD paradigms explained in the paper. Interestingly, the authors also explore the combination of different distillation paradigms. The method proposed was also convincing, obtaining superior performance over the baselines with a compressed model. The ablation section is thorough, showing that each part of the final method contributed to the performance.

A minor criticism is that the study was only done on CenterPoint based models, and that there were no point-cloud based models evaluated (i.e., the conclusions were for pillar- and voxel-based inputs only). It would be interesting to see how point-based models (such as PointRCNN) would compress and hold up to these conclusions.

Miscellaneous:
L82 vs. L 85: "Suppl." vs "Appendix" term mismatch

---

> ### Author Response · Authors · 2022-08-02
> **Response to Comments from Reviewer nWAU**
>
> Thanks for your thoughtful feedback! We are really glad for your appreciation to our paper. We provide responses to specific questions as below.
>
>
> **Q1: The study was only done on CenterPoint-based models, and there were no point-cloud based models evaluated.**
>
> Thanks for your constructive feedback. We agree that including point-based methods would make our work more general and comprehensive. Actually, we also investigate point-based architecture at the initial of this work.
>
> (1) However, point-based methods achieve sub-optimal performance on practical ring-view datasets (i.e. datasets including LiDAR point clouds in all 360 degrees among ego-vehicle) such as Waymo, Lyft, nuScenes, Argoverse, etc. As shown in the following table, the state-of-the-art point-based detector IA-SSD [50] performs inferior than voxel/pillar-based detectors on Waymo.
> | Method | Type | LEVEL2 mAPH |
> | :------: | :----: | :----: |
> | CP-Voxel | Voxel-based | 65.58 |
> | CP-Pillar | Pillar-based | 61.56 |
> | IA-SSD [50] | Point-based | 58.08 |
>
> (2) Besides, the efficiency of point-based detectors decreases quadratically when the number of points increases linearly, as the points-wise feature encoding network (e.g. PointNet++) relies on furthest point sampling with $O(n^2)$ time complexity. On the contrary, voxel/pillar-based detectors are less sensitive to the number of points with voxelization operation. Therefore, point-based methods are rarely employed in the practical ring-view datasets with dense point clouds.
>
> (3) To further verify the generality of our approach, we try our method on KITTI with an anchor-based detector -- SECOND [46] (see our response to Q2 of Reviewer Sm5s). Experimental results show that our compression conclusions are still valid and KD methods also show outstanding performance on the new dataset as well as the new detector.
> In addition, we are glad to make our work more general by constructing investigations on point-based detectors on more suitable datasets (e.g. KITTI). However, due to the large work overhead and limited time during the rebuttal, we leave it to future research. Once we obtain comprehensive results, we will update them to the paper.
>
>
>
> **Q2: Miscellaneous: L82 vs. L 85: "Suppl." vs "Appendix" term mismatch.**
>
> Thanks for your careful reading and pointing that out. We will fix this error in the revision.
>
>
> **Q3: In Section 5.3, it is observed that the student model can outperform the teacher model. Is there intuition for why this may be? Intuitively, unless the student model has access to labels, it should not have enough information to correct these mistakes.**
>
> Thank you for your comments.
> (1) Actually, we use GT labels during the KD process, following the default setting for previous KD methods. As shown in the ''Label KD part'' of Sec. 4.1 (see line 235), the teacher assisted GT set $\hat{y}^{\text{KD}}$ is constructed by combining GT labels $y$ and confident teacher predictions $\hat{y}^t$. Besides, for KD methods other than label KD, student models still leverage GT labels as naive supervised training. In this regard, student models always have access to GT labels and have enough guidance to correct teacher's mistakes. We will add clarification of this part in the revision.
>
> (2) We also provide an extra experiment to investigate how teacher prediction and GT labels influence the performance of Label KD. As shown in the following table, although the student can achieve reasonable performance with only teacher prediction as supervision, it still needs GT labels to obtain gains compared to the no distillation model. We will include these results in the revision.
>
> | Detector | GT | Teacher Pred | LEVEL2 mAPH |
> | :------: | :----: | :----: |  :----: |
> | CP-Pillar-v0.48 | $\surd$ | $\times$ | 56.24 |
> | CP-Pillar-v0.48 | $\times$ | $\surd$ | 54.66 |
> | CP-Pillar-v0.48 | $\surd$ | $\surd$ | 57.54 |

---

> > ### Comment · Reviewer_nWAU · 2022-08-09
> > **Post-rebuttal**
> >
> > Thank you for your thoughtful responses. I am happy with the paper, and look forward to the final version.

---

> > > ### Author Response · Authors · 2022-08-09
> > > **Thank you**
> > >
> > > Thanks for your great effort in helping us to strengthen the paper. We are so humbled to receive such recognition. We will punctiliously revise our paper to include the above discussions and experiments in the final version.

---

### Official Review · Reviewer_Qi4p · 2022-07-12

**Rating:** 7
**Confidence:** 5
**Soundness:** 3 good
**Presentation:** 3 good
**Contribution:** 3 good

**Summary:**

This paper focuses on the task of efficient 3D object detection. It first studies how to obtain student models with good trade off between accuracy and efficiency. Then, it proposes an improved KD pipeline incorporating an enhanced logit KD method and a teacher-guided student model initialization to facilitate transferring teacher model’s feature extraction ability to students through weight inheritance. Extensive experiments on Waymo dataset show the efficiency of the proposed method.

**Questions:**

Please see the comments above.

**Limitations:**

Yes, the limitations have been discussed in the supplementary file.

**Strengths And Weaknesses:**

$\textbf{Strength}$

1. The motivation of solid, and the paper is well organized.

2. Extensive experiments are conducted to analyze the designs of efficient student networks and the performance of benchmark knowledge distillation for 3D object detection.

3. Experiments on WOD demonstrate the effectiveness of the improved knowledge distillation.

$\textbf{Weakness}$

1. The designs of student network are limited to input resolution compression or model width/length compression. And each variant only involves in one compression method. Discussions on more designs of lightweight student network are encouraged, to see if the conclusions are still held. It will also increase the difficulty of knowledge distillation.

2. In Tab. 3, what is the performance of applying all three kinds of KDs (like Eq. 4). It seems that only the result of each KD (i.e., logit KD, feature KD and label KD) has been analyzed.

---

> ### Author Response · Authors · 2022-08-02
> **Response to Comments from Reviewer Qi4p**
>
> Thanks for your constructive reviews and questions. We are encouraged by your appreciation to our contribution. We provide responses to specific questions as below.
>
>
> **Q1: Discussions on more designs of lightweight student networks are encouraged, to see if the conclusions are still held. It will also increase the difficulty of knowledge distillation.**
>
> Thanks for your thoughtful comments.
> (1) As our paper focuses on providing the first systematic study for efficient 3D detection with KD, we investigate the most widely-adopted and easy-to-use compression schemes including input resolution and model width/depth compression. We have not tried sophisticated layer-wise compression or combined different compression methods which require complicated algorithms such as neural architecture search in EfficientNet [40] and is beyond the study of this paper. In fact, we have discussed such limitations in the ''Limitations'' part of supplemental materials, which will be our future work.
>
> (2) Here, we still would like to provide some experimental results to combine model and resolution compression along with some KD attempts. As shown in the following table, we further apply model width compression based on CP-PP-v0.4. The obtained CP-PP (b) and CP-PP (c) both achieve higher CPR compared to CP-PP-v0.4, because coarser-resolution detectors are supposed to have more architecture-level redundancy with less input information. These results also support our claim in the supplemental materials (see line 152-156 in the supplemental material). Furthermore, by comparing CP-PP (b) and CP-PP (c), we can find that our compression conclusion ''PFE has less redundancy to be reduced'' is still valid.
>
> | Detector | Width-PBE | Width-BEF |Width-Head | Voxel Size (m) | Params (M) |  Flops (G) | Acts (M) | Latency (ms) | mAP@R40 | CPR |
> | :-: |:-: | :-: | :-: | :-: | :-: | :-: | :-: | :-: | :-: | :-: |
> | CP-PP | 1.00 | 1.00 | 1.0 | 0.32 | 5.2 | 333.9 | 303.0 | 157.9 | 59.09 | - |
> | CP-PP-v0.4 | 1.00  | 1.00  | 1.00  | **0.40** | 5.2 | 212.9 | 197.7 | 103.4 | 57.55 | 0.64 |
> | CP-PP (a) | 1.00 | **0.875** | **0.875** | 0.32 | 4.0 | 260.1 | 267.7 | 134.7 | 58.53 | 0.54 |
> | CP-PP (b) | 1.00 | **0.875** | **0.875** | **0.40** | 4.0 | 163.9  | 175.5 | 92.1 | 57.36 | 0.67 |
> | CP-PP (c) | **0.875** | **0.875** | **0.875** | **0.40** | 4.0 | 163.2 | 173.2 | 91.1 | 56.92 | 0.66 |
>
> (3) In addition, we also provide extra results of applying various KD methods to the newly designed CP-PP (b). As shown in the following table, although CP-PP (b) has similar no distillation results with CP-PP-v0.4, it obtains fewer benefits from distillation among all KD strategies. These results illustrate that more sophisticated compression can increase the difficulty of KD.
>
> | Detector | No Distill | KD | GID-L |  FitNet | Mimic | FG |GID-F | Label KD | Ours | Flops(G) | Acts (M) |
> | :-: | :-: | :-: | :-: | :-: | :-: | :-: | :-: | :-: | :-: | :-: | :-: |
> | CP-PP | 59.09 | - | - | - | - | - | - | - | - | 333.9 | 303.0 |
> | CP-PP-v0.4 | 57.55 | 57.51 | 57.54 | 57.89 | 58.57 | 58.44 | 58.26 | 58.10  | 59.24 | 212.9 | 197.7 |
> | CP-PP (b) | 57.36 | 56.93 | 56.70 | 57.15 |  57.81 | 57.48 | 57.77 | 57.57 | 58.62  | 163.9 | 175.5 |
>
>
> **Q2: In Tab. 3, what is the performance of applying all three kinds of KDs (like Eq. 4). It seems that only the result of each KD (i.e., logit KD, feature KD and label KD) has been analyzed.**
>
> Thanks for your comments. We provide the combination results of three KD methods in Tab. 4 based on CP-Voxel-XXS along with the synergy effect analysis. Furthermore, we show the synergy effect results of all designed architectures in Table S1 in the supplemental material due to the page limit of the main paper. Hope these results could address this concern.

---

> ### Author Response · Authors · 2022-08-09
> **Looking forward to further discussion**
>
> Thank you for your constructive comments and suggestions. If you have other questions and concerns, please let us know and we are happy to further discuss. Thank you again for your time.

---

### Official Review · Reviewer_Sm5s · 2022-07-12

**Rating:** 6
**Confidence:** 5
**Soundness:** 3 good
**Presentation:** 3 good
**Contribution:** 3 good

**Summary:**

This paper proposes to use knowledge distillation to train more lightweight 3d detectors (which tend to suffer from heavy computational overhead). For given teacher networks (CenterPoint Pillar and Voxel variants), the authors experimentally determine a collection of student networks (e.g. via reducing width/depth, input size in certain ways) that achieve good efficiency and performance.

Then given these student architectures, the authors experiment with different knowledge distillation schemes (including a proposed scheme - called Pivotal Position Logit KD - that puts more emphasis on foreground regions). There are a few main findings, but one punchline (across a number of experiments mostly done on the Waymo dataset) is that the authors are able to realize a 3d detection model that outperforms the teacher model with ~2x less FLOPS.


**Questions:**

* For the proposed CPR (cost performance ratio) - why not simply use actual measured runtime instead of activations?
* In the section on selecting student architectures, it’s not clear to me how these models are trained — what knowledge distillation approach was used, what dataset etc?
* A suggestion for Table 3: have a more descriptive caption --- explain that the “Ours” column is not re-explained until a later section
* On pivotal position logit KD
    * The authors mention “instance-aware local region imitation” in Sec 4.2 but don’t really talk about it until Sec 5.1 (at which point it’s called something else — pivotal position logit KD).  So am I correct that these are referring to the same idea?
    * It still makes sense to include some background contribution in the loss function — how is this incorporated in pivotal position logit KD?
    * Focal loss is often used in 2d detection as a way to deal with foreground/background imbalance — could this idea be adapted for knowledge distillation for 3d detection?


**Limitations:**

Yes

**Strengths And Weaknesses:**

The clear strength of this paper is that it is able to get a clear win over CenterPoint with the Waymo dataset and the paper also offers insights that would help others that would like to do knowledge distillation for a similar architecture.

On the other hand, the ideas in the paper are incremental — mostly this paper can be viewed as an empirical exploration of existing knowledge distillation ideas (with the exception of the pivotal position KD idea — but even the high level idea here of emphasizing foreground regions is not novel).  There is a running theme that some KD approaches are synergistic with each other — and these are discovered empirically, but there are no principles offered by which a reader can think about this at a higher level.  Finally, it is hard to know how general these findings are.  For example, how do we know that the choice of student models does not depend on the specifics of each KD approach?  And can we expect any of these findings to generalize beyond CenterPoint based models?  Would they generalize to non-Waymo datasets?

---

> ### Author Response · Authors · 2022-08-02
> **Response to Comments from Reviewer Sm5s (3/3)**
>
>
> **Q6: For PP Logit KD: (1) It is referring to the idea: instance-aware local region imitation. (2) How can it include some background contribution in the loss function? (3) Focal loss is often used in 2d detection. Can focal loss work on KD for 3D detection?**
>
> (1) *Comparison with instance-aware local region imitation*: Actually, we have claimed that PP logit KD is motivated by the imbalance of foreground and background imbalance issue and previous designs in the 2D area to alleviate this problem (see line 246-250 and 271-272). And we have compared the difference between PP logit KD and previous instance-wise imitation methods in line 51-65 of the supplemental material.
>
> As a 3D detector needs to make predictions on a large $150m \times 150m \times 6m$ 3D space, the foreground and background imbalance issue is more extreme than 2D detection with just a single image. Besides, a single position on the BEV pillar image can cover a large region of $0.8m \times 0.8m \times 6m$. Thus, this extreme sparsity and informative BEV detection paradigm pose a new challenge to knowledge distillation for 3D object detection. To this end, our teacher-guided pivotal position selection offers sparser selected areas, focusing on more informative areas (see visualization comparison between PP KD and instance-aware KD: https://drive.google.com/file/d/1B6wMRke_Ivy7broikGvXKPCHy7sXEJhR/view?usp=sharing). This design is tailored to 3D object detection.
> Our experimental results also demonstrate that the proposed PP logit KD is more powerful with only $\frac{1}{5} \sim \frac{1}{20}$ attention regions compared to 2D instance-aware local region imitation methods (see Table 3 and Table 7 in the main paper as well as Table S1 in the supplemental material).
>
> Besides, how to select better local regions for imitation is a long-standing research problem in KD for 2D detection areas such as [22, 42, 6, G] from 2017 to present. In this regard, leveraging local region imitation to solve foreground/background imitation is a common motivation and a research direction which is non-trivial and very important to advancing this area.
>
>
> (2) *Background areas*: As our confidence and rank PP logit KD relies on teacher prediction to select pivotal positions, if the background points are predicted with high or top-ranked confidence, our PP logit KD will also apply distillation loss on those background positions. As for the loss function, PP logit KD just set the corresponding positions of $m_{\text{cls}}$ in Eq. (2) to one, which then makes distillation loss active in those positions.
>
> (3) *Focal loss*: As we don't know exactly what the focal loss you are referring to, we discuss both the two scenarios of applying focal loss: ($i$)focal loss on GT supervision loss; ($ii$) focal loss on distillation objective.
>
> Scenario ($i$):  Focal loss is a defacto selection in 3D object detection to solve foreground/background imbalance and is already equipped in the supervised training objective in all our trained models.
>
> Scenario ($ii$): As far as we know, focal loss is not widely employed as a distillation loss for 2D object detection as shown in Mimicking [22], FG [42], FGD [G], etc. Still, we implement a focal distillation loss similar to the supervised loss. The experimental results are shown in the following table. Our PP logit KD is around 0.7\% higher than focal loss on CP-Voxel-XS. As for CP-Pillar-v0.64, since the capability difference between teacher and student are large, focal loss even suffers performance degradation compared to vanilla KD, while our PP logit KD consistently brings performance boost.
>
> |   Detector  | Role | No distill | KD | Focal loss | PP Logit KD |
> | :-: | :-: | :-: | :-: | :-: | :-: |
> |  CP-Voxel    | Teacher | 64.29 |  - | - | - |
> |  CP-Voxel-XS | Student | 62.23 | 62.81 | 63.48 | 64.16 |
> |  CP-Pillar    | Teacher | 59.09 |  - | - | - |
> |  CP-Pillar-v0.64 | Student | 52.81  | 50.78 | 46.11 | 54.32 |
>
> The reason for the inferior performance of focal loss for distillation is that it will emphasize regions that are most different among teacher and student but not necessarily be information-rich areas. Instead, such prediction difference might be caused by the capability gap between teacher and student. Hence, emphasis on such regions might be a suboptimal strategy and even penalize student learning. We will include these results in the supplemental materials in the revision.
>
>
> [G] Focal and Global Knowledge Distillation for Detectors.

---

> ### Author Response · Authors · 2022-08-02
> **Response to Comments from Reviewer Sm5s (2/3)**
>
>
> **Q3: Why not simply use measured runtime instead of activations for CPR?**
>
> Thank you for the comments. Actually, we have discussed the reason for using activations rather than runtime/latency in both the evaluation metrics part of the main paper (see line 129-131) and supplementary material (see Section S3.1 in the supplemental material).
> The main reason lies in the fact that the runtime of a detector largely depends on the hardware devices and operation-level optimizations. Besides, the runtime is even not stable on the same machine with different machine statuses such as temperature. Experimental results and discussions can be found in Section S3.1 of the supplemental material. Since the used hardware devices and operation optimization largely vary between different research groups, we use a machine-independent metric -- activations to calculate CPR to benefit more future research.
>
> **Q4: For Section 3: it’s not clear to me how these models are trained — what KD approach was used, what dataset etc?**
>
> Thank you for your comment. We clarify it as below:
>
> (1) In Section 3, our objective is to investigate how to design an efficient 3D detector, where we simply train the designed detectors without any knowledge distillation methods as the training schema in OpenPCDet [41] (see line 124-125). We will further add a clarification: this part is agnostic to KD methods in the revised paper.
>
> (2) For the dataset, we train those models on Waymo Open Dataset with 20\% training samples, which is also the default training schema of OpenPCDet [41] on WOD. Related clarifications can be found in: line 78, line 122-123, line 131 as well as the table header of Table 1 and Table 2 as LEVEL 2 mAPH is the specific metric of WOD.
>
>
> **Q5: Table 3: explain that the ''Ours'' column is not re-explained until a later section.**
>
> Thank you for pointing out such a confusing expression. We will add an explanation of ''Ours'' in the caption of Table 3 to make it easier to understand in the revision.

---

> ### Author Response · Authors · 2022-08-02
> **Response to Comments from Reviewer Sm5s (1/3)**
>
> Thank you for your reviews. We provide responses to specific questions as below.
>
>
> **Q1: The ideas in the paper are incremental. The high-level idea of pivotal position KD is also not novel.**
>
> (1) Thanks for your comment. Note that our paper focuses more on *exploring the potential of knowledge distillation for efficient 3D detectors*, which aims to provide a general model-agnostic solution to obtain efficient while well-performed 3D detectors and encourage future research to obtain more efficient 3D detectors by improving the compression strategies or KD manners. In addition, our conclusion drawn from both extensive detector compression investigations and KD benchmark results can benefit future research in the community. In fact, our technical contribution -- improved KD pipeline only takes a small part of this paper, and we hope the reviewer can also pay attention to our other contributions.
>
> (2) *Clarification about the novelty of PP logit KD*: As this question is also asked more concretely in Q6, please refer to our answer to Q6.
>
>
> **Q2: The Generality of the approach.**
>
> (1) *Generality of the selection of student model*: Note that the training and selection of students are agnostic with KD methods. We develop CPR to choose students that have a good trade-off between performance and efficiency. When selecting student models, we consider their CPRs and try to cover a wide range of model capabilities for validating the generality and scaling ability of KD methods. Please note that experimental results presented in the S3.2 of supplementary material demonstrate that CPR correlates well with student models' accuracy and efficiency. Besides, our experiments in the semantic segmentation task further demonstrate the generality of CPR as a comprehensive criterion to assess student models (see the response to Q2 of Reviewer hUXR).
>
> (2) *Why CenterPoint-based model and Waymo dataset*: We focus on the CenterPoint-based method, as its different variants rank 2nd on Waymo [E] and 1st in nuScenes [F], respectively, which demonstrates that it is a top-performing model at the time of submission. Besides, our experiments are constructed on WOD as it is the largest annotated public 3D detection dataset, around $5 \sim 15 \times$ larger than other 3D detection datasets such as nuScenes, Lyft, Argoverse and KITTI. We believe that KD methods verified on top-performed detectors and large-scale dataset should be more general and beneficial for both future research and industrial applications.
>
> (3) *Generality on another detector and dataset*: To further verify the generalization of our KD methods on both detector-level and dataset-level, we provide the model compression and KD results for KITTI dataset based on the anchor-based detector: SECOND [46]. As shown in the following tables, both our compression conclusion and KD methods can generalize to the new detector and dataset, where SECOND (d) surpasses teacher performance by around 0.5\% with $3.5\times$ fewer flops. We will add these results to the revised paper.
>
> | Detector | Width-PBE | Width-BEF | Params (M) |  Flops (G) | Acts (M) | Latency (ms) | mAP@R40 | CPR |
> | :-: |:-: | :-: | :-: | :-: | :-: | :-: | :-: | :-: |
> | SECOND | 1.00 | 1.00 | 5.3 | 80.5 | 69.3 | 77.4 | 67.24 | - |
> | SECOND (a) | 1.00 | **0.50** | 2.0 | 26.0 | 40.0 | 54.3 | 66.64 | 0.70 |
> | SECOND (b) | **0.50** | 1.00 | 4.6 | 72.4 | 65.2 | 70.6 | 65.70 | 0.50 |
> | SECOND (c) | **0.50** | **0.50** | 1.4 | 20.5 | 35.9 | 46.1 | 64.21 | 0.68 |
> | SECOND (d) | **0.75** | **0.50** | 1.6 | 23.0 | 38.0 | 51.8 | 65.62 | 0.69 |
>
>
> | Detector | No Distill | KD | GID-L |  FitNet | Mimic | FG |GID-F | Label KD | Ours | Flops(G) | Acts (M) |
> | :-: | :-: | :-: | :-: | :-: | :-: | :-: | :-: |  :-: | :-: | :-: | :-: |
> | SECOND | 67.24 | - | - | - | - | - | - | - | - | 80.5 | 69.3 |
> | SECOND (d) | 65.62 | 66.06 | 66.34 | 66.00 | 66.37 | 66.58 | 66.75 | 67.03 | 67.70  | 23.0 | 38.0 |
>
> (4) *Generality on new semantic segmentation task*: We also verified the effectiveness of our proposed KD methods on the 3D semantic segmentation task, which strongly demonstrates the generalization ability of our proposed method (see the answer to Q2 of Reviewer hUXR).
>
> [E] https://waymo.com/open/challenges/2021/real-time-3d-prediction/
>
> [F] Scaling up Kernels in 3D CNNs.

---

> ### Author Response · Authors · 2022-08-09
> **Looking forward to further discussion**
>
> Thank you for your constructive comments and suggestions. If you have other questions and concerns, please let us know and we are happy to further discuss. Thank you again for your time.

---

### Official Review · Reviewer_hUXR · 2022-07-12

**Rating:** 6
**Confidence:** 4
**Soundness:** 3 good
**Presentation:** 3 good
**Contribution:** 3 good

**Summary:**

This paper studies knowledge distillation for point cloud object detection. Designing efficient architecture to process large scale point clouds is a long-standing challenge for autonomous driving. To that end, this paper tries to improve model efficiency by leveraging knowledge distillation. This paper conducts an extensive study of existing knowledge distillation methods under several settings. It proposes a new metric called Cost Performance Ratio (CPR) to measure the performance and efficiency trade-off. Also, this paper provides two systematical ways to compress both inputs and architectures respectively. In addition, this paper analyzes the feature responses in both teacher network and student network and proposes a novel way to select a sparse set of locations in the feature map to distill. The best performing method using the proposed knowledge distillation achieves very strong performance on the Waymo dataset and even outperforms its teacher counterpart while only requiring less than half of teacher flops.

**Questions:**

I would appreciate feedback/comments to the following questions:

1. What do authors think of sparse/dense distillation? If we work on sparse models such as Object DGCNN and 3DETR, can we still do similar distillation as this paper does?

2. Can authors provide code to facilitate reproducibility? I think there are many details/hyperparameters still missing. It is not straightforward to reproduce the work.

**Limitations:**

I don't have concerns about the potential negative societal impact.

**Strengths And Weaknesses:**

Overall, I think this paper makes good contributions to point cloud knowledge distillation. It provides an adequate study of existing knowledge distillation methods under 3D object detection settings. In addition, it introduces a novel metric to compare different knowledge distillation methods. Moreover, the analysis of feature response leads to a module to select a sparse set of features to distill. I summarize strengths and weaknesses as follows.

Strengths:

1. First, I like the adequate study of existing distillation methods conducted in the paper. These experiments show a clear overview of how knowledge distillation works in the point cloud object detection scenario. Also, the reimplementations of existing methods (e.g., PV-RCNN++) are faithful and very close to SOTA models.

2. The proposed CPR is technically sound. This metric considers both cost and performance and can be potentially used in general knowledge distillation settings. Also, this metric reflects empirical results/observations of models.

3. The analysis of feature response is interesting and intriguing. Based on the feature response, this paper comes up with a way to select features to distill rather than distilling the whole feature map to avoid focusing on feature noises.

4. Empirically, the proposed method shows quite strong performance compared to existing knowledge distillation baselines. Its best performing student network even outperforms the teacher counterpart while only requiring 44% of teacher flops. What is more convincing is that the improvements are consist under almost all settings.

Weaknesses:

1. My first concern is about the generalizability of the proposed method. It seems that this paper focuses on two specific architectures -- PointPillars/SparseConvs and PVRCNN. I am curious that whether this proposed method is applicable to Transformer-based model such as Object DGCNN and 3DETR. As those methods are already sparse, can the proposed sparse distillation still be applied in those cases?

2. I understand this paper focuses on object detection, which is a coarse prediction task compared to fine-grained prediction like semantic segmentation. So I am wondering, if this method is useful for dense tasks like semantic segmentation or flow prediction. My worry is that for dense prediction tasks, the feature selection module won't introduce too much improvement compared to dense feature map distillation. It would be great if authors can provide evidences on tasks other than object detection.

---

> ### Author Response · Authors · 2022-08-02
> **Response to Comments from Reviewer hUXR (2/2)**
>
> **Q2: Whether this method is useful for dense tasks like semantic segmentation or flow prediction?**
>
> (1) Thanks for your constructive and interesting question. We agree that dense prediction tasks such as semantic segmentation requires fine-grained supervisions and might hinder the effectiveness of our sparse distillation strategy (i.e. pivotal position KD). However, we argue that since the student model already has dense GTs as supervision in training, dense distillation loss on massive uninformative points and regions, such as road points, might be redundant and can overwhelm the overall distillation loss. Instead, our sparse distillation might help the student focus on more important areas by using teacher prediction as regularization.
>
> (2) Here, we follow the design principle of PP logit KD and adapt it to handle the dense semantic segmentation task. We apply distillation loss on points with predictions that are correct but less confident than the teacher. Our simple design is motivated by three intuitions: ($i$) Points that are correctly predicted with lower confidence are often some challenging cases that the model is struggling but also has the capability to handle. By harvesting knowledge from a high-performing teacher model, the student can learn to match the confidence level of the teacher which provides more information than the one-hot GT. ($ii$) Points that are correctly predicted with higher confidence are often easy samples that have very close prediction confidence to the teacher model. Considering that these samples are already handled well by the model, they have low chance to benefit from distillation but might cause redundancies. ($iii$) Points that are incorrectly predicted by the student are often cases that might be out of the ability of student models.
>
> Note that we are only able to verify this simple design following the intuition of our sparse distillation strategy in this rebuttal period. Dedicated designs might further strengthen the results.
> Specifically, we have the confidence of student predictions $\text{conf}^s$, the confidence of teacher predictions $\text{conf}^t$ and a pre-defined threshold $\tau$. We will only apply distillation loss for student predictions that are correct and have $\text{conf}^s + \tau < \text{conf}^t$.
>
> (3) We also provide experimental results for our design on the 3D semantic segmentation dataset ScanNet. Here, we use a small version of MinkowskiNet [D] for fast verification. As shown in the following table, first, we try both model width and input resolution compression to obtain student models, and select MinkowskiNet14-v0.04 as the student model for KD due to its higher CPR. Then, we validate logit KD, our PP logit KD and our TGI on it. Both our proposed PP logit KD and TGI obtain improvements. In particular, our sparse PP logit KD surpasses the dense logit KD method with around 0.8\% gains.
> Our statistics also show that our PP logit KD only leverages 19.03% points for distillation at the first epoch and 3.66% points for distillation at the last epoch. These experiments and statistics demonstrate that sparse distillation can also work on the dense prediction task. We will include the above results in the supplemental material.
>
>
> | Model | Width | Voxel Size (m) | Params (M)| Flops（T） | Acts (M) | mIoU | CPR|
> |:-:|:-:|:-:|:-:|:-:|:-:| :-:| :-:|
> |MinkowskiNet14 (teacher)| 1.0 | 0.02 | 1.7|46.2|27.9| 65.77 | -|
> |MinkowskiNet14-w0.5| **0.5** | 0.02 | 0.5 | 18.2 | 17.4| 61.84 |0.60|
> |MinkowskiNet14-v0.04| 1.0 | **0.04** | 1.7 | 5.7 | 8.9 | 62.82 |0.78|
>
> | Model | Role | No distill | Logit KD | PP Logit KD | TGI | Flops (T) | Acts (M) |
> |:-:|:-:|:-:|:-:|:-:|:-:|:-:| :-:|
> |MinkowskiNet14| Teacher | 65.77 |-|-|-| 46.2 | 27.9 |
> |MinkowskiNet14-v0.04| Student | 62.82 | 63.65 | 64.40 | 64.22 | 5.7 | 8.9 |
>
>
> [D] 4D Spatio-Temporal ConvNets: Minkowski Convolutional Neural Networks
>
> **Q3: What do authors think of sparse/dense distillation? Can we still do similar distillation on sparse models?**
>
> Thank you for your comments.
>
> (1) *Sparse/dense distillation*: In our opinion, as GT labels can provide necessary supervision for training the neural network, dense distillation could waste attention on some uninformative positions that could even overwhelm the distillation loss on pivotal positions. On the contrary, sparse distillation can only focus on informative regions/positions, regularizing the training objective of student networks to those informative and improvable regions/positions with teacher guidance.
> Our experimental results on 3D detection and 3D semantic segmentation tasks also support that sparse distillation can be a stronger strategy than dense distillation.
>
> (2) *Similar distillation on sparse models*: As this question is similar to Q1, please refer to our answer to Q1.
>
>
> **Q4: Can authors provide code to facilitate reproducibility?**
>
> As the claim in the abstract, we promise that we will make our code publicly available once accepted.

---

> > ### Comment · Reviewer_hUXR · 2022-08-08
> > **Thank you for the response**
> >
> > I thank authors for providing insightful comments to my questions. These additional experiments are very promising especially the dense prediction task. My concerns are addressed and I recommend acceptance of this paper. I encourage authors to include these discussions/results in the paper and add the sparse distillation results in the next revision. I believe they will further make this paper stronger.

---

> > > ### Author Response · Authors · 2022-08-08
> > > **Thank you**
> > >
> > > We sincerely appreciate your effort in helping us to strengthen the paper and your support for our work! We will punctiliously revise the paper based on the above discussion and experimental results in the next revision. They can definitely make our paper more solid!

---

> ### Author Response · Authors · 2022-08-02
> **Response to Comments from Reviewer hUXR (1/2)**
>
> Thank you for your thoughtful reviews. We are grateful for your appreciation and interesting questions. We provide responses to specific questions as below.
>
> **Q1: Whether this proposed sparse distillation still be applicable to sparse Transformer-based models?**
>
> Thank you for your thoughtful question, which is actually an open question and an under-studied problem.
> From our perspective, sparse distillation is still applicable for sparse transformer-based detectors such as DETR [A], Deformable DETR [B], Object DGCNN [C], etc.
>
> (1) As for sparse transformer-based detectors that directly provide instance predictions, their predictions actually rely on learning to some sparser reference points and corresponding position features. For example, each object query in Deformable DETR or Object DGCNN is decoded into a reference point and $K$ neighboring points in order to focus only on those most informative positions.
>
> (2) Although sparse detectors can directly provide sparse instance predictions, we argue that our sparse distillation (i.e. pivotal position KD) focuses on sparser and more fine-grained position-level information (see visualization comparison between PP KD and instance-level KD: https://drive.google.com/file/d/1B6wMRke_Ivy7broikGvXKPCHy7sXEJhR/view?usp=sharing). In this regard, it should still be applicable to sparse models with some specific modifications.
>
> (3) Here, we take Object DGCNN [C] as an example and provide two possible sparse distillation designs:
>
> (3.1) As the transformer encoder and decoder of Object DGCNN are similar to Deformable DETR [B], it can be simply extended to a two-stage variant as Deformable DETR. In the two-stage variant, the transformer encoder will regard each pixel as an object query and construct a dense prediction on it. Top scoring positions are picked as reference points. This is similar to our designed rank PP KD which enforces the student to imitate the prediction of teacher top-rank positions. Therefore, we can directly apply our sparse rank PP KD to those dense scoring predictions between teacher and student. Besides, we will also carry on feature imitation on those teacher top-ranked positions between teacher and student.
>
>
> (3.2) As for the one-stage variant of sparse detectors, learnable object queries will be decoded into reference points and neighboring points, so the sparse distillation can be constructed on those points and their corresponding BEV features. Specifically, we can first match the positive object queries of teacher and student as query pairs by checking whether they are matched to the same GT box. Then, we can enforce the decoded reference and neighboring points of the student to mimic their paired teacher counterparts. Besides, we will construct imitation on BEV features of those reference and neighboring positions between teacher and student.
>
>
> (4) We are trying to construct experimental verifications about our above designs based on Object DGCNN. However, since we need to change to new dataset (i.e. nuScenes), new codebase (i.e. MMDetection3D) and new detection paradigm (transformer-based) with limited time and resources, we have not obtained results now. We will keep attempts and update them in the comments or revised version once we get results.
>
>
> [A] End-to-end object detection with transformers.
>
> [B] Deformable detr: Deformable transformers for end-to-end object detection.
>
> [C] Object dgcnn: 3d object detection using dynamic graphs.

---

### Meta-Review · Area_Chair_fN4U · 2022-08-24

**Recommendation:** Accept
**Confidence:** Less certain

**Metareview:**

In this paper, the authors propose a new method for knowledge distillation for 3D object detection in point cloud data. This problem is quite important for self-driving cars and 3D computer vision. The goal of their work is to compress models to achieve reasonable trade-offs in compute performance versus accuracy. The authors explore these questions using two popular forms of 3D detection: pillar-based and voxel-based architectures (a) and focus on extensive experimentation with the Waymo Open Dataset.

The authors first examined how to build student-teacher models with good trade-offs between accuracy and computational demand, introducing a new metric the Cost Performance Ratio (CPR). The authors then systematically explore a series of knowledge distillation methods (e.g.  logit KD, label KD, and teacher-guided initialization) to identify their best model. The end result of their search is to identify a student model that is able to outperform the teacher model but with ~2x less FLOPS.

The reviewers commented positively on the strength experiments and baselines as well as the selection and CPR metric. The main issues surfaced by the reviewers focused on the generalizability of the results outside of 3D object detection including whether the methods or results may be ported to other 3D detection architectures or other localization tasks. The authors responded with some discussion and new early experiments highlighting that the work may be ported to problems in semantic segmentation.

From my perspective, the paper comes across as technically sound with strong experiments and a solid overall result. I am concerned about the generality of these results. As this work largely focuses on 3D object detection of specific detection architectures, I could imagine that this this work would be better suited for conferences geared towards the topics of 3D object detection and self-driving cars (e.g. CVPR and associated workshops). In that spirit, I would consider this paper borderline. However, because this work shows promise for other architectures and problems, I view this work as potentially having more generality. It would be incumbent on the authors to revise their manuscript accordingly to include additional discussion on this generality as well as showcase encouraging results on other domains. This paper will be conditionally accepted assuming all of these changes are made to this manuscript.

(a) Note that this work seems to exclude the recently popular range-based methods [1] and it would be important for the authors to add discussion to their paper accordingly. E.g.

[1] LaserNet: An Efficient Probabilistic 3D Object Detector for Autonomous Driving
Gregory P. Meyer, Ankit Laddha, Eric Kee, Carlos Vallespi-Gonzalez, Carl K. Wellington (2019)


**Award:**

No

---

### Decision · Program_Chairs · 2022-09-14

Accept